



# High-frequency NO₃⁻ isotope ($\delta^{15}$N, $\delta^{18}$O) patterns in groundwater recharge reveal that short-term land use and climatic changes influence nitrate contamination trends

Martin Suchy[1], Leonard I. Wassenaar[2], Gwyn Graham[1], and Bernie Zebarth[3]

*Correspondence to*: Leonard. I. Wassenaar (l.wassenaar@iaea.org)

[1] Environment and Climate Change Canada, 201-401 Burrard St., Vancouver, BC, Canada
[2] Present address: International Atomic Energy Agency, Vienna International Centre, 1400, Vienna, Austria
(l.wassenaar@iaea.org)
[3] Fredericton Research and Development Center, Agriculture and Agri-Food Canada, 850 Lincoln Rd., PO Box
20280 Fredericton, NB, Canada E3B 4Z7

**Abstract.** Poultry manure is the primary source of nitrate (NO₃⁻) exceedances in the transboundary Abbotsford-Sumas aquifer (Canada-USA) based on synoptic surveys two decades apart, but serious questions remained about seasonal and spatial aspects of agricultural nitrate fluxes to the aquifer to help better focus remediation efforts. We conducted over 700 monthly $\delta^{15}$N and $\delta^{18}$O of nitrate assays, focusing on newly recharged groundwater (<5 yr.-old) over a five-year period to gain new insight on spatiotemporal sources and controls of groundwater nitrate contamination. NO₃⁻ concentrations in recharge ranged from 1.3 to 99 mg N L⁻¹ (*n*=1041) with a mean of 16.2 ±0.4 mg N L⁻¹. These high-frequency isotope data allowed us to identify 3 distinctive nitrate flux patterns, i) nitrate in recharge influenced by synthetic fertilizer inputs ii) nitrate in recharge impacted by short-term climatic and local agricultural crop rotations and iii) long-term widespread manure and synthetic fertilizer inputs. A key finding was that the source(s) of nitrate in recharge could be quickly influenced by short-term near-field management practices and stochastic climatic factors, which linger and ultimately impact long-term nitrate contamination trends. Overall, the isotope data affirmed a subtle decadal-scale shift in agricultural practices from manure towards fertilizer nitrate sources, nevertheless poultry-derived N remains a predominant source of nitrate contamination. Because the aquifer does not support denitrification, remediation of the Abbotsford-Sumas aquifer is possible only if agricultural N sources are seriously curtailed, a difficult proposition due to longstanding high-value intensive poultry and berry operations over the aquifer.



## 1 Introduction

The global widespread use and over-application of synthetic and manure N-nutrients in agriculture has caused widespread groundwater nitrate ($NO_3^-$) contamination in numerous aquifers around the world (Hasleur et al., 2005; Hamilton and Helsel, 1995; Spalding and Exner, 1993). Furthermore, with global trends towards increased agricultural intensification, threats to surface and groundwater quality are correspondingly heightened (Vorosmarty et al. 2000; Böhlke, 2002). In agricultural settings, elevated shallow groundwater $NO_3^-$ concentrations typically result from a combination of inappropriate animal manure or synthetic fertilizer over-applications, incomplete nitrogen uptake by crops, and/or from elevated residual soil organic nitrogen in the non-growing season (Canter, 1997). The risk of $NO_3^-$ contamination is especially high in phreatic aquifers with coarse grained permeable soils and minimal propensity for natural attenuation and remediation processes, such as microbial denitrification. Studies have used nitrate isotopes ($\delta^{15}N$, $\delta^{18}O$) to investigate the sources of nitrate (Mitchell et al., 2003; Wassenaar et al., 2006; Xue et al., 2009), while others have used isotopes to examine the history and fate of groundwater nitrate (Böhlke et al., 1995; Kellman and Hillaire-Marcel, 2003). Others used nitrate isotopes to assess soil N transformations (Savard et al., 2010), or temporal variations in agricultural leachate to groundwater (Ostrom et al., 1998; Loo et al. 2017; Savard et al., 2007).

Concentrations of non-agricultural $NO_3^-$ in aquifers that are low (<1 mg N L$^{-1}$) and below drinking water standards can usually be attributed to sources like wet or dry atmospheric N deposition, organic N from plant decomposition or land breakage, and geological sources that are mobilized due to disruptions in water recharge fluxes such as commencement of irrigation (Canter, 1997). Choi et al. (2003) reports when groundwater $NO_3^-$ concentrations are consistently below 3 mg N L$^{-1}$ with $\delta^{15}N$ values between +5 and +8 ‰, then soil organic N (average $\delta^{15}N$ +5 ‰) is likely to be a primary source. Loo et al. (2017) reported non-agricultural soil $\delta^{15}N$ nitrate ranges of +3.7 to +4.9 ‰ (Table 1).

Sources of nitrate from animal waste arise from dispersed agricultural field applications and/or point-source manure storage facilities (liquid and solid). Under aerobic soil conditions, $NO_3^-$ quickly forms from oxidation of $NH_4^+$ after manure application (Aravena et al, 1993). Due to preferential volatilization of $^{14}N$ in gaseous $NH_3$ from $NH_4^+$ during wet storage and/or application of manure, manure-derived $NO_3^-$ is accordingly enriched in $^{15}N$ (Kendall 1998). Nitrate $\delta^{15}N$ from manures typically ranges between +10 to +20 ‰, (Wassenaar, 1995; Kreitler, 1975), while $\delta^{15}N$ values from domestic septic waste range between +10 to +25 ‰ (Heaton, 1986; Aravena and Robertson, 1998), generally revealing little $^{15}N$ isotopic resolution between these two waste sources. Poultry manure has average $\delta^{15}N$ values of approximately +7.9 ‰ in the study area (Loo et al., 2017; Wassenaar, 1995). In North America, Urea ($CO(NH_2)_2$) (46-0-0), is one of the most common forms of synthetic fertilizers used (Overdahl et al., 2007). Other forms of synthetic fertilizers include ammonium-nitrate ($NH_4-NO_3$) (34-0-0) and ammonium-sulfate ($NH_4-SO_4$) (22-0-0). Each of these are



manufactured by fixation of atmospheric N ($\delta^{15}$N = 0 ‰), resulting in $\delta^{15}$N values from -2.8 to +0.3 ‰. In the
Abbotsford-Sumas aquifer area, berry-specific fertilizer blends are commonplace (Table 1), where N is derived
from one of the above sources (Loo et al., 2017; Wassenaar, 1995). The $\delta^{18}$O values of synthetic fertilizer
derived $NO_3^-$ typically range between +18 to +22 ‰, because the oxygen in nitrate originates from air $O_2$
($\delta^{18}$O = +23.5 ‰) and $^{18}$O depleted $H_2O$ (Amberger and Schmidt, 1987). Nitrate derived from $NH_4$-$NO_3$
fertilizers, where 50 % of the oxygen is from nitrification of $NH_4$ fertilizer and 50 % is from synthetic $NO_3^-$
fertilizer, have reported $\delta^{18}$O values around +13 ‰ (Aravena et al, 1993).
In the phreatic transboundary Abbotsford-Sumas (ASA) (Canada-USA, Figure 1), long-term nitrate
contamination trends and isotopic studies have been conducted over several decades. The isotopic
apportionment of $NO_3^-$ sources in the aquifer was based on two, decades apart, synoptic nitrate isotopic
sampling that revealed that poultry manure was the predominant source of groundwater $NO_3^-$, with long-term
shifts towards inorganic fertilizer sources (Wassenaar, 1995, Wassenaar et al., 2006) due to changes in
agricultural practices (Zebarth et al, 2015). One critique of the previous synoptic nitrate isotope efforts was that
sampling (and hence interpretations) was biased to summer 'snapshots', and thereby could be biased, especially
for the numerous shallow and highly responsive water table wells spanning the aquifer and the winter-biased
recharge. The seasonal dynamics of $NO_3^-$ sources and fluxes and the potential for isotopic changes due to soil
and unsaturated zone $NO_3^-$ cycling were not evaluated, and need to be considered to improve surface nutrient
applications and agricultural management practices.
To address this knowledge gap, we conducted high-frequency (monthly) $NO_3^-$ concentration and
isotope sampling of the ASA over a 5-year period, with a focus on water table wells having residence times of
<5 years as determined by $^3$H-He age dating. Our aim was to determine whether high-frequency (monthly)
isotope nitrate and isotope ($\delta^{15}$N, $\delta^{18}$O) assays improved previous interpretations of sources and process, and
whether important seasonal changes in the proportion of $NO_3^-$ sources recharging to groundwater were
overlooked by occasional synoptic snapshots. Our goal was to gain improved insight on the spatiotemporal
sources and controls of groundwater-nitrate dynamics, and thereby to help better inform agricultural nutrient
management practices and potential $NO_3^-$ remediation efforts in the aquifer.
**2 Materials and Methods**
**2.1    Study Area and Hydrogeologic Setting**
The Abbotsford-Sumas aquifer is a shallow phreatic transboundary aquifer located in southwestern British
Columbia, Canada, and northwestern Washington State, USA (Figure 1). The ASA is the most intensively
studied nitrate-contaminated aquifer in Canada (Zebarth, 1998, 2015), and covers an area of about 200 km$^2$,
with approximately 40 % of the surface area in Canada (Cox and Kahle, 1999). Our study area encompassed
approximately 40 km$^2$ on the Canadian side of the aquifer, between the Abbotsford International Airport and
the Canada-USA border (Figure 1). Land use on the aquifer is predominantly commercial raspberry and





blueberry production, mixed with intensive commercial poultry barn operations (Figure 1) and is <5 % rural
residential; unpublished data (BC Ministry of Agriculture).

The aquifer is typically 10-25 m thick, but reaches 70 m thickness in the south-east part of the aquifer

(Cox and Kahle, 1999). The aquifer comprises coarse glacio-fluvial sand and gravel with minor till and clayey
silt lenses (Armstrong et al., 1965), with glacio-marine clays confining the aquifer below (Halstead, 1986). The
high sand and gravel content results in a high transmittance of water, with mean hydraulic conductivities (K) of
$1.6 \times 10^{-3}$ m s$^{-1}$ (Chesnaux et al. 2007) to $9.5 \times 10^{-4}$ m s$^{-1}$ (Cox and Kahle, 1999). The thin surface soils (0-70 cm)
are medium-textured aeolian deposits, moderately-well to well-drained, and are classified as Orthic Humo-
Ferric Podzols (Luttermerding, 1980).

Average annual precipitation across the aquifer (1981-2010) is 1538 mm, of which 70 % falls between

October and March (Environment Canada, 2014). Annual recharge estimates range from 850 to 1100 mm
(Zebarth et al, 2015), and water table depths typically vary between 2 to 20 m below surface depending on the
location and season. Annual water table fluctuations average ~3.6 m (Scibek and Allen, 2006). The overall flow
direction in the aquifer is south (Figure 1), southeast, and southwest at linear velocities of up to 450 m yr$^{-1}$
(Liebscher et al., 1992; Cox and Kahle, 1999).

The aquifer is highly vulnerable to surface derived $NO_3^-$ and other contamination because of i)

intensive agricultural activity, ii) the highly permeable soil, coarse sand and gravel lithology and iii) high
precipitation amounts in the fall and winter when nutrient uptake by crops is lowest and $NO_3^-$ leaching
potential is highest (Kohut et al., 1989; Liebscher et al. 1992). Elevated groundwater-nitrate concentrations
exceeding drinking water guidelines are observed since the 1970's (Zebarth et al. 2015). Mitchell et al. (2003)
and others (Wassenaar et al., 1995) showed vertical stratification of nitrate was linked to agricultural practices,
with highest nitrate concentrations (>20 mg N L$^{-1}$) occurring in shallow water table regions (<10 mbgs), while
average groundwater-nitrate concentrations in deep wells were lower and relatively stable over time. Based on
Environment and Climate Change Canada (ECCC) monitoring, the highest seasonal and temporal variations in
$NO_3^-$ are found in wells screened near the water table. Both seasonal and long-term temporal variations in
groundwater-nitrate over decadal timeframes are well documented (Liebscher et al. 1992; Graham et al., 2015).
The aquifer has little widespread intrinsic capacity to sustain microbial denitrification (self-remediation) because
of largely aerobic conditions and the low organic content of the aquifer materials (Wassenaar, 1995), but it can
occur in localized pockets.
**2.2 Sample Collection and Analysis**
Monthly groundwater samples (*n*=56 per well) were collected from 19 selected monitoring wells from
September 2008 to March 2013. These wells were selected based on the following criteria: 1) ground water
having a <5-year residence time based on $^3$H-He age-dating (Wassenaar et al., 2006); 2) representative spatial
coverage within the monitoring network; and 3) aerobic wells where denitrification does not occur (Tesoriero,



2000; Wassenaar et al., 2006). These criteria helped to ensure that high-frequency nitrate and isotopic patterns
stem from short-term nitrate responses unaffected by historical or subsurface biogeochemical processes or
mixing with deeper water, and could therefore be more explicitly linked to contemporary landscape and
agricultural activities and practices happening roughly within a 5-year timeframe.

Static water level measurements were taken prior to pumping and were reported in meters above mean

sea level (masl). Groundwater was sampled from the wells using a Grundfos® stainless steel submersible pump,
Teflon® lined LDPE tubing, and stainless-steel fittings and valves. Well water was pumped through a flow-
through cell housing a calibrated YSI® multi-probe sonde (temperature, pH, specific conductance, oxidation
reduction potential (ORP), and dissolved oxygen (DO)). General chemistry, and $NO_3^-$ isotope water samples
were collected after at least three well volumes were purged and the YSI® field parameters were stabilized. All
bottles were rinsed 3x with sample water prior to filling. Water samples for major ion and nutrient
concentrations were taken in 1 L LDPE bottles, filtered through 0.45 μm cellulose acetate membrane filters,
stored at 5°C and analyzed within 5 days for nitrate using standard ion chromatography techniques. Nitrate
concentrations were determined at the Pacific-Yukon Laboratory for Environmental Testing in North
Vancouver, BC, Canada. Nitrate results are reported as mg N $L^{-1}$.

Samples for nitrate isotope analyses ($\delta^{15}N$, $\delta^{18}O$) were field filtered through 0.45 μm cellulose acetate

membrane filters and frozen (-40 °C) in 125 ml HDPE bottles. Nitrate isotope assays were conducted by the
University of Calgary Stable Isotope Laboratory, using microbial reduction to $N_2O$ described elsewhere
(Casciotti et al., 2002; Sigman et al., 2001). All $\delta^{15}N$ values are reported relative to the atmospheric air reference
(Mariotti, 1983) and normalized by analyzing reference materials IAEA-N3 ($\delta^{15}N_{AIR}$ = +4.7 ‰), USGS32
($\delta^{15}N_{AIR}$ = +180 ‰), USGS34 ($\delta^{15}N_{AIR}$ = -1.8 ‰), USGS35 ($\delta^{15}N_{AIR}$ = +2.7 ‰) along with samples. The
analytical uncertainty for $\delta^{15}N$ was ±0.5 ‰. The $\delta^{18}O$ values were reported relative to the VSMOW reference
(Coplen, 1994) and determined by analyzing reference materials IAEA-N3 ($\delta^{18}O_{VSMOW}$ = +25.6 ‰), USGS32
($\delta^{18}O_{VSMOW}$ = +25.7 ‰), USGS34 ($\delta^{18}O_{VSMOW}$ = -27.9 ‰), and USGS35 ($\delta^{18}O_{VSMOW}$ = +57.5 ‰). The analytical
uncertainty for $\delta^{18}O$ was ±1.0 ‰.

Nitrate and chloride concentrations were log-transformed prior to analysis to ensure normal

distributions and were evaluated using Principal Component Analysis (PCA) and Factor Analysis. Statistical
analyses (at the 95 % confidence level), including multivariate time series analyses were conducted using the
Kruskall-Walis methods for determining seasonality, log-normal transformations, Mann-Kendall trend analyses
and Gaussian mixture and Bayesian clustering models using WQHydro®, ProUCL 5® and XLSTAT®
(Lettenmaier, 1988; Thas et al., 1998). Seasonal Mann-Kendall trend analysis were deemed inappropriate for
evaluating nitrate seasonality as the repeating periods were correlated to precipitation patterns instead of



calendar month, and because peak nitrate concentration timings varied from year to year, resulting in a
determination of non-seasonality.
**3 Results and Discussion**
**3.1    Groundwater Nitrate Chemistry**
Results of monthly nitrate concentrations in the water table wells in the aquifer over the 5-year sampling period
ranged from 1.3 to 99.0 mg N L$^{-1}$ (*n*=1041), having a mean concentration (± SE) of 16.2 ±0.1 mg N L$^{-1}$.
Approximately 76 % of the shallow groundwater locations (16 of 19 sites) exceeded the maximum allowable
concentration (MAC) of 10 mg N L$^{-1}$ in the Canadian Drinking Water Guidelines (Health Canada). These
nitrate exceedances were consistent with previous observations of high nitrate concentrations in shallow wells
in the aquifer (Hii et al., 1999). Previous studies reported $NO_3^-$ concentrations exceeding the MAC in 58 %, 69
% and 59 % of wells (Wassenaar 1995, Zebarth et al., 1998, Wassenaar et al., 2006), respectively. The current
study only had a ~50 % well overlap with previous investigations because early studies also sampled deeper
monitoring wells containing older groundwater.
A time-series analysis showed that overall $NO_3^-$ concentrations steadily increased in the targeted
shallow wells over our 5-year study period, which contrasted with long-term declines observed for a wider
depth variety of wells in the Canadian portion of the aquifer (Zebarth et al., 2015). Graham et al. (*2015*)
identified several key drivers causing the short-term (intra- and inter-year) nitrate trends (increases or declines)
that contrasted with the long-term (inter-decadal) declines. These key drivers were primarily stochastic rainfall
patterns (wet vs. dry years) and short-term land-use change factors. The overall increasing nitrate trend in the
19 wells could be attributed to the marked increases in $NO_3^-$ concentrations in three of the wells occurring in
the second half of our study. These nitrate increases were attributed to i) clearing of an adjacent woodlot, ii)
application of large quantities of poultry manure as a soil amender to the cleared land up-gradient of PC-25 and
PC-35 in 2011, and iii) a raspberry field up-gradient of US-02 that underwent a renovation cycle (described in
Zebarth et al., 2015) which likely also included soil N amendments. Wells 94Q-14, PA-25 and PA-35 did not
exceed the nitrate MAC because these sites were located up-gradient of the most intense agricultural
production areas.
Almost half the 19 shallow monitoring wells (47 %) showed $NO_3^-$ seasonality, with maximum
concentrations usually occurring in the springtime. Nitrate accumulates in the soil and root zones over the
summer, and a large proportion of nitrate flushing to the water table happens with the first major recharge
events in the rainy season (Kowalenko, 2000). Subsequent recharge typically has lower nitrate concentrations as
the availability of dissolved soil nitrate drops. Previous evidence of $NO_3^-$ flushing in the fall is shown by
Wassenaar (1995) and Zebarth et al. (1998), when precipitation, recharge rates, and soil-$NO_3^-$ are at their peak.
Coupled with vadose zone infiltration lag-times of several months (Herod et al., 2015), accordingly peak $NO_3^-$
concentrations reaching the water table are observed in the springtime.




All wells were aerobic, with DO levels usually > 3 mg L$^{-1}$ (Supplementary Table), however, two sites
(ABB-03 and US-02) showed a short intervals of lower DO levels (<1 mg L$^{-1}$) in the winter months, coinciding
with higher water tables. Chloride levels were on average 8.7 ± 3.0 mg L$^{-1}$. At 6 sites (91-10, 91-15, PA-25, PA-
35, US-02, and US-05), $NO_3^-$ and Cl concentrations exhibited a covariance (Pearson's R correlation coefficients
>0.5), suggesting similar sources. Three sites (PC-25, FT5-12 and FT5-25), exhibited variability between $NO_3^-$
and Cl, however, the Cl peaks usually lagged behind $NO_3^-$ peaks by 1-3 months, which was surprising
considering Cl$^-$ is considered a conservative tracer, although this was also seen by Malekani (2012). The
remaining sites exhibited limited seasonal nitrate and chloride variability or correlation.
**3.2    Nitrate N and O Isotopes**
Overall, the mean (±SE) nitrate $\delta^{15}N$ value for the 19 study wells was +7.9 ±.11 ‰ (*n*=717), which was
consistent with $\delta^{15}N$ values of local poultry manure sources (Wassenaar, 1995; Loo et al., 2017) as summarized
in Table 1. Mean nitrate $\delta^{18}O$ was -1.7 ±0.06 ‰ (*n*=717), which was typical of values derived during the
nitrification of manure or synthetic fertilizers (Xue et al., 2009). Previously measured groundwater $\delta^{18}O_{H2O}$ in
the aquifer ranged narrowly between -10 to -12 ‰ and coupled with O derived from air (+23.5 ‰), the current
nitrate $\delta^{18}O$ values were comparable with earlier $\delta^{18}O$ values (Wassenaar 1995) of -1.0 ± 0.26 ‰ (*n*=16) and
+0.5 ± 0.79 ‰ (*n*=40) a decade later (Wassenaar et al., 2006).
To further assess sources and seasonality of nitrate in these 19 shallow wells, the results were evaluated
using nitrate concentrations and isotopic compositions. A Keeling plot of $1/NO_3^-$ vs $\delta^{15}N$ (Figure 2a),
supported by a Gaussian mixing model suggests three main nitrate groupings with the following proportions
and interpretation: i) a historical mixing (47 %) trend between high $NO_3^-$ and high $\delta^{15}N$ (manure-derived) and
low $NO_3^-$ and low $\delta^{15}N$ values (fertilizer-derived), ii) fertilizer and soil N dominant (47 %) low $NO_3$ and low
$\delta^{15}N$ (+2 to +4‰), iii) intermediate $NO_3^-$ and mid $\delta^{15}N$ (+8 ‰), mixed source of manure/soil N/fertilizer (6
%). A Bayesian VVV (Volume, Shape and Orientation) clustering model using $\delta^{15}N$ and $NO_3^-$ suggested 5
possible groupings (Figure 2b), with means shown in Table 2. These findings altogether suggest that field-scale
agricultural management practices up-gradient of the monitoring wells resulted in at least 4 quantifiably
distinctive nitrate isotopic clusters (Table 3 - Source Grouping).
Another clustering approach, based on $\delta^{15}N$ trends and seasonality in the 19 wells over the course of
the study was also evaluated. In this case, sites were separated into 4 clusters (Table 3 - Trend Grouping) as
follows: A) No trend with stable $\delta^{15}N$ values (SD < ±1.0 ‰); B) No trend with variable $\delta^{15}N$ values (SD> ±1.0
‰); C) $^{15}N$ enrichment trends; D) $^{15}N$ depletion trends.





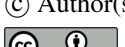

### 3.3.1 Nitrate Isotopic Variations

Considering the Bayesian and Gaussian clustering approaches together, we separated the nitrate and isotope data into 4 distinctive groups based on their isotopic values (3 primary groups and 1 sub-group), both in relation to each other and to well-known $NO_3^-$ sources.

Group 1a was impacted by synthetic fertilizer and/or residual soil N and showed little isotopic variability, while Group 1b was similar but impacted by clear short-term spikes in $\delta^{15}N$ and $NO_3^-$. Group 2 was dominated by poultry manure with some influence of $^{15}N$ depleted sources, while Group 3 was dominated solely by poultry manure N.

The four wells categorized into Group 1a, with $\delta^{15}N$ values of +3 to +8‰ representing 21 % of the 19 sites (PA-25, PA-35, 91-07, and US-04), had a mean $\delta^{15}N$ value of +5.0 ‰. The isotope distribution of these samples suggests they were dominated by synthetic fertilizers and natural (background) soil N sources ($\delta^{15}N$ of -1.0 ‰ and +4.0 ‰, respectively). Loo et al. (2017) reported that weighted $\delta^{15}N$ of fertilizer treatment leachate in the ASA is +3.2 ± 2.3 ‰. Sampling wells in this group did not exhibit large seasonal swings in $NO_3^-$ concentration or $\delta^{15}N$ values, although strong seasonality was found for $NO_3^-$ in wells PA-25 and PA-35. These isotope data suggest a combination of annual synthetic fertilizer applications with occasional poultry manure application as a soil amendment, which is a common agricultural practice in this area, particularly with blueberry crops.

The Group 1b wells were distinctive because the mean nitrate $\delta^{15}N$ value was more negative than poultry manure (+6.7‰, like Group 1a values), but spanned a wider $\delta^{15}N$ range from +2 to +16‰, representing 11 % of the wells (PC-25 and PC-35). In addition, both exhibited nitrate $^{18}O$ enrichment, coupled with increasing $\delta^{15}N$ values (Figure 3A) and $NO_3^-$ concentrations. Well PC-25 was possibly subjected to localized or temporal soil zone denitrification since some $\delta^{18}O$ values increased above +5 ‰, however, groundwater DO values were never below 8.8 mg $L^{-1}$, suggesting microbial denitrification process were unlikely in this well. The positive $\delta^{15}N$ values coupled with elevated $NO_3^-$ (Figure 3B) concentrations were more likely the result of soil amendment practices whereby poultry manure is applied to fields during crop replacement cycles to augment soil carbon and nitrogen content (Zebarth et al., 2015). As previously indicated, this site may also have been affected by recent adjacent woodlot clearing and poultry manure application following planting of a new blueberry crop in 2011-2012. If the elevated $\delta^{15}N$ after January 2012 are omitted from these two wells, the mean $\delta^{15}N$ drops to +4.2 ‰, which corresponds to Group 1a. Furthermore, most of the Group 1a/1b wells fall along the same groundwater flow path (Figures 1 and 4).

Wells categorized as Group 2 had a mean $\delta^{15}N$ of +7.8 ‰, which corresponded to both manure leachate (+7.3 ± 1.2 ‰; Loo et al., 2017) and poultry manure in general. The more $^{15}N$ depleted samples were likely influenced by synthetic fertilizers or residual soil N, while $^{15}N$ enriched samples represented temporal soil



zone denitrification. Group 2 wells include: 91-03, 91-15, 94Q-14, ABB-02, ABB-03, ABB-05, FT5-12, FT5-25,
PB-20 and PB-35. Wells in this group were in the majority, representing 53 % of the sites, and as with Group 1
did not exhibit large seasonal or inter-annual swings in $NO_3^-$ concentrations or their $\delta^{15}N$ values, other than
both $NO_3^-$ concentrations and $\delta^{15}N$ values were more elevated compared to Group 1. Based on these results, it
appeared that poultry manure applications, or excess residual soil N from historical poultry manure applications
influenced these wells.

The Group 3 wells (91-10, US-02 and US-05) had a mean $\delta^{15}N$ value of +12.6 ‰, which was more

enriched in $^{15}N$ than local poultry manure or manure leachates (Table 1). These $^{15}N$ enriched results likely
resulted from ammonia volatilization of the source poultry manure and temporal soil zone denitrification.
Ammonia volatilization occurs in poultry manure piles and during field application of wet manure. The
mineralized residual ammonium can have $\delta^{15}N$ values up to +25 ‰, but is dependent on pH, temperature,
humidity and other environmental factors (Kendall, 1998). Group 3 sites are all located down-gradient of
current and former poultry barns or known locations of on-field poultry storage piles, which was shown by
Wassenaar (1995) to result in isotopically enriched $\delta^{15}N$ values in soil N from +7.5 to +13.6 ‰ that are flushed
to the aquifer.

### 3.3.2   5-Year Isotopic Trends

The 19 monitoring wells were evaluated based on their nitrate $\delta^{15}N$ and $\delta^{18}O$ isotopic trends over the

study period. The trend evaluation was conducted using Mann-Kendall (monthly data) and Seasonal Kendall
(bi-monthly data) non-parametric tests for detection of upward or downward trends in a time series at the
$p > 0.05$ level of significance. For individual wells, if there was insufficient evidence to detect a trend, individual
well results were grouped as being 'stable' or 'variable', depending on whether the $\delta^{15}N$ standard deviation was
< or > 1.0 ‰, respectively. Wells exhibiting seasonality were identified as Group B. The analysis showed no
significant temporal trend in $\delta^{15}N$ during the study period, however, if results from the three nitrate 'spiking'
sites (US-02, PC-25 and PC-35) were removed, a significant overall $\delta^{15}N$ depletion trend was observed. This
finding corresponded to the previously reported finding of a decadal-scale nitrate $^{15}N$ depletion trend in the
aquifer, which was attributed to a long-term shift from manure to fertilizer use (Wassenaar et al., 2006).

Four wells (91-15, ABB-02, ABB-05 and FT5-12) were classified into Trend Group A, where analyses

did not support a significant upward or downward $\delta^{15}N$ trend and ±SD ≤1.0 ‰ (Figure 5A). All four wells
(21%) were from Distribution Group 2, where $\delta^{15}N$ were +6 to +10 ‰. Interestingly, all Group A sites
exhibited appreciable $NO_3^-$ variability, but only FT5-12 depicted any seasonality, with peak nitrate
concentrations occurring in winter, likely the result of soil N mobilization following higher precipitation
periods. Average $NO_3^-$ concentrations were 16.1± 6.4 mg N L$^{-1}$. The de-coupling of $\delta^{15}N$ from $NO_3^-$





suggested a consistent isotopic $NO_3^-$ source, with no microbial transformations, whose concentrations were
likely driven by seasonal periods of enhanced recharge.

Trend Group B comprised 6 wells (91-10, PA-25, PA-35, PB-20, US-02 and US-05) with no

significant $\delta^{15}N$ trend over the study period (Figure 5B), but exhibiting high $\delta^{15}N$ variability around the mean
($\pm$SD $\geq$1.0 ‰). The degree of $\delta^{15}N$ and $NO_3^-$ variability differed for most wells in this group; however, all
sites exhibited strong $\delta^{15}N$ and $NO_3^-$ coupling, with at least a 5 ‰ change in $\delta^{15}N$ and 15 mg N $L^{-1}$ fluctuation
in $NO_3^-$ concentrations. In US-02, decreasing DO concentrations were associated with decreasing $\delta^{15}N$
values; however, in this case $NO_3^-$ and Cl concentrations were correlated, suggesting fertilizer loading was the
cause (Supplementary Table). In fact, the up-gradient field of this well had undergone a renovation cycle in
the preceding months, where old raspberry plants were removed followed by application of poultry manure to
the field prior to replanting. It should be noted that Cl is common in synthetic fertilizer, but was
undocumented if fertilizers were applied to the up-gradient field. Sites 91-10 and US-05 showed similar $\delta^{15}N$
and $NO_3^-$ fluctuations, albeit smaller in magnitude, with corresponding increases in chloride and elevated
dissolved oxygen concentrations. Sites 91-10 and US-05 are close to each another (<200 m apart) along a
similar groundwater flow path, suggesting these variations are linked. No other sites in this group were
spatially proximal. Sites PB-20, PA-25 and PA-35 exhibited varying degree of coupled $\delta^{15}N$ and $NO_3^-$
seasonality, suggesting nitrate leaching was the primary driver of $NO_3^-$ variability. For PA-25, increasing $NO_3^-$
concentrations with $\delta^{15}N$ enrichment (although variable in degree) were systematically observed each winter,
suggesting nitrate mobilization occurred during peak winter rainfall periods.

Six sites were identified as Trend Group C, with increasing $\delta^{15}N$ trends (91-03, 91-07, FT5-25, PC-25,

PC-35, and US-04). These sites were evenly distributed between Distribution Groups 1a (3) 1b (2) and 2 (1),
suggesting one driver controlling local $NO_3^-$ concentrations and $\delta^{15}N$ values. Enriching $^{15}N$ trends (often
along a flow path) are usually associated with progressive microbial denitrification, however, all sites had high
DO aerobic concentrations (>5 mg $L^{-1}$). Sites PC-25 and PC-35, which exhibited some degree of coupled $^{15}N$
and $^{18}O$ enrichment at a 2:1 ratio, also showed increasing $NO_3^-$ concentrations, suggesting heavy loading of
poultry manure. Prior to the marked increase of $NO_3^-$ and $\delta^{15}N$ in the spring of 2012, PC-25 and PC-35
exhibited a significant, albeit gradual, increasing trend (Figure 5C). This revealed a second subtle driver – the
increased precipitation that occurred between 2008-2011 (Environment and Climate Change Canada, 2014),
and its effect on groundwater nitrate concentrations, as shown by Graham et al. (2015). Wells 91-03, FT5-25,
and US-04 did not undergo any up-gradient crop replacement or soil amendments, and exhibited various
degrees of $NO_3^-$ and $\delta^{15}N$ seasonality, further strengthening the climatic link as a potential driver. The
increasing $\delta^{15}N$ trend could be linked to the enhanced mobilization and infiltration of $^{15}N$ depleted soil-N
where $^{14}N$ nitrogen was preferably volatilized.





Group D exhibited a $^{15}$N depletion trend (Figure 5D), and consisted of monitoring wells 94Q-14, PB-
35 and ABB-03, and had a negative $\delta^{15}$N shift of 1-3 ‰, and $\delta^{15}$N values between +6 to +10 ‰ (Group 2).
Well 94Q-14 showed $\delta^{15}$N seasonality, but not in $NO_3^-$, with concentrations mostly below the MAC. PB-35
showed small seasonality in $NO_3^-$ concentrations but none in $\delta^{15}$N, indicating possible mixing and dilution
due to a shift in nitrogen sources. Wassenaar et al., 2006 suggested that a negative $\delta^{15}$N shift may be attributed
to the longer-term change in nitrogen sources used from poultry manure to synthetic fertilizers. Lastly, ABB-
03 showed no significant trend in $NO_3^-$ concentrations or in $\delta^{18}$O, however, $\delta^{15}$N and $\delta^{18}$O were correlated,
while $\delta^{15}$N and $NO_3^-$ were inversely correlated. Furthermore, ABB-03 exhibited short intervals of anaerobic
conditions that corresponded to periods of $^{15}$N enrichment and decreasing $NO_3$, suggesting localized
denitrification, which were repeatable to various degrees on a seasonal basis, but was most prominent in 2011.
These findings suggest localized and temporally limited denitrification may be occurring in the soil root zone
in some areas, contributing to $^{15}$N enrichment and variability of $NO_3^-$ concentrations. Site ABB-03 was not
near Fishtrap Creek (Figure 1), which Tesoriero (2000) and (Wassenaar et al., 2006) identified as a localized
denitrification hot spot. Depletion in $^{15}$N at these sites appeared to be from temporal drivers that could be
overlooked in one-time synoptic sampling (Wassenaar, 1995).
**4 Conclusions and Outlook**
This study represents an unprecedented high-frequency 5-year seasonal spatiotemporal study of water
table well with over 700 nitrate isotopic assays, revealing the dynamics of nitrate recharging the transboundary
Abbotsford-Sumas aquifer. The high (monthly) temporal frequency of nitrate and isotopic data aimed to
address concerns that infrequent nitrate isotopic or concentration synoptic samplings of shallow ground water
overlooks important factors of seasonality that may be key drivers of nitrate sources and fluxes to shallow
aquifers. Indeed, our study revealed new important scientific information not previously seen in the synoptic
surveys that will help managers better tackle nutrient management strategies to help reduce ground water
pollution.
Overall, and unsurprisingly, we found the predominant perennial source of nitrate to the aquifer at all
spatiotemporal scales within the 5-year intensive sampling period was animal waste (poultry) sources, which
was already known for decades. Nitrate concentrations in young (<5 yr.-old) and newly recharged groundwater
was persistently high in nitrate, ranging from 1.3 to 99 mg-N L$^{-1}$, with a mean of 16.2 mg-N L$^{-1}$, and well in
exceedance of the Canadian drinking water MAC of 10 mg-N L$^{-1}$ for 76 % of the wells. The study also verified
a postulated and subtle decadal-scale shift towards $^{15}$N depleted nitrate sources, likely reflecting systematic
changes in agriculture practices from the early days of indiscriminate manure disposal towards more targeted
use of synthetic fertilizers, or from changes in crop types and associated nutrient practices, as evidenced by the
mean $\delta^{15}$N value for nitrate of +7.9 ± 3.0 ‰ compared to +10.2 ± 4.0 ‰ in the 1990s. Synthetic fertilizer and



soil N are a comparatively higher N loading in the central portions of the ASA, but is flanked on both sides by
higher poultry manure dominated N loadings. The high nitrate concentrations in contemporary recharging
groundwater means widespread nitrate contamination of the aquifer is likely to persist into the foreseeable
future, and our data affirm little evidence for persistent or widespread attenuation of nitrate by subsurface
denitrification processes, at any time of the year. Nitrate remediation of the aquifer will only be possible if
agricultural N sources are dramatically reduced or eliminated, which is unlikely to be an acceptable proposition
if the inter-generational high-value poultry and commercial blueberry and raspberry crops are at stake.

In some wells we found that localized agricultural practices (i.e. N soil amendment) had a nearly

immediate multi-year negative impact, mainly exhibited by marked increases of poultry-derived N, and lasting
for several years across the seasons.  This common practice resulted in spatial clustering and differing short-
term trends for water table nitrate and isotopes across the aquifer (Figure 4 and 6), further revealing that
infiltrating $NO_3^-$ and its isotopic composition can change quickly in direct response to contemporary near-field
practices. Conversely, this suggests N source cutoff as a remediation effort could be similarly as effective.
Despite 53 % of shallow wells showing no isotopic trends, 47 % showed an isotopic enrichment or depletion
trend, and about half of the wells exhibited nitrate seasonality in $NO_3^-$ concentrations and/or $\delta^{15}N$ values
controlled by temporal infiltration of residual mineralized N or weak, short-term denitrification.

Due to the rapid shift in $NO_3^-$ and isotopic values of recharging groundwater immediately following

field renovation and soil amendment practices, this study reinforces the importance of designing and
conducting appropriate spatio-temporal nitrate sampling to reduce the risk of misinterpreting nitrate and
isotopic data though the more common practice of occasional synoptic surveys. The dynamics of nitrate in
younger (<5 yr.-old) water table wells, however, also imply it would be prudent to monitor deeper, older
groundwater which smooth out short-term fluctuations and hence record longer-term and aquifer-wide
trends.

For the ASA agricultural area specifically, measuring the impact of changes in nutrient management

practices associated with the switch from raspberry to blueberry crops or field renovation is required to
determine its impacts on groundwater nitrate dynamics. Decisions on future aquifer nitrate management need
to take into consideration permanent or cyclical changes in the planned crop types, and the associated nutrient
management practices involved with them. Subtle shifts in nitrate in the ASA may be unexpectedly influenced
by the recent increased planting of blueberries in place of raspberries, which appear to be less reliant of
cyclical poultry manure soil amendments.



## Acknowledgements

Funding for this project was provided by Environment Canada (LIW, MS, GG), Agriculture and Agri-Food Canada (BZ), and the Canadian Water Network. The authors would also like to thank Steven Taylor from the Isotope Lab at the University of Calgary. Geoff Koehler assisted with sampling. Tommy Diep assisted with GIS mapping and field activities (Environment and Climate Change Canada).

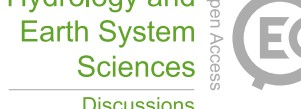

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





**Figure 1**: Location of the Abbotsford-Sumas aquifer (ASA), southwestern B.C., Canada and northwestern
Washington State, USA, along with simplified agricultural land-use and sampling locations with ground water
mean residence times (MRT) of < 5 years. Arrows show the approximate groundwater flow direction.

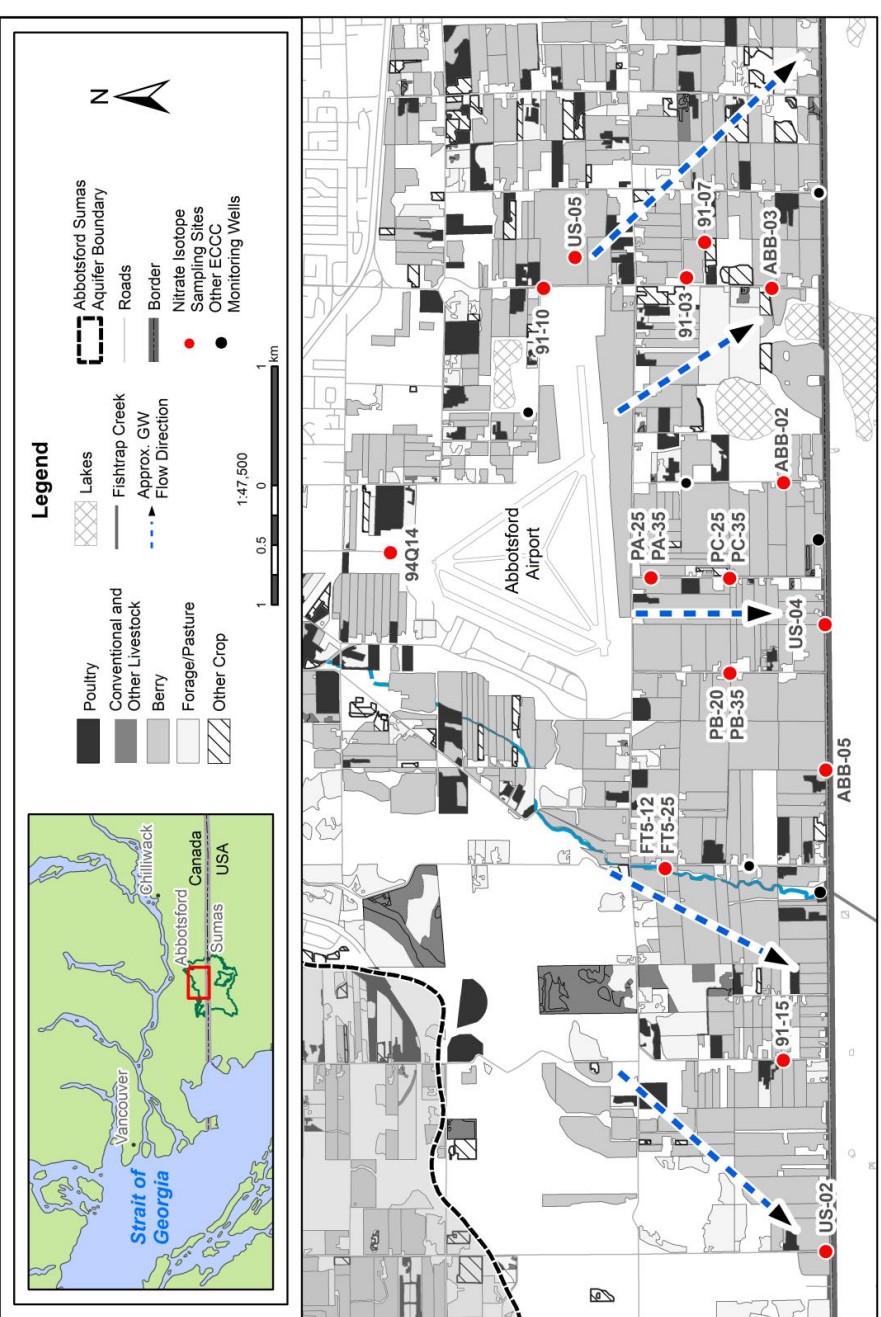






**Figure 2**: a) Keeling plot of $1/NO_3$ (x-axis) vs. $\delta^{15}N$ (y-axis). Three distinct groups are i) Arrow represents mixing line between fertilizer and manure endmembers, ii) (Green) wide range $NO_3$ (mineral fertilizer), iii) (Purple) middle group (manure/fertilizer mixture). b) $\delta^{15}N$ vs. Nitrate Bayesian clustering model suggest 5 distinct groupings.

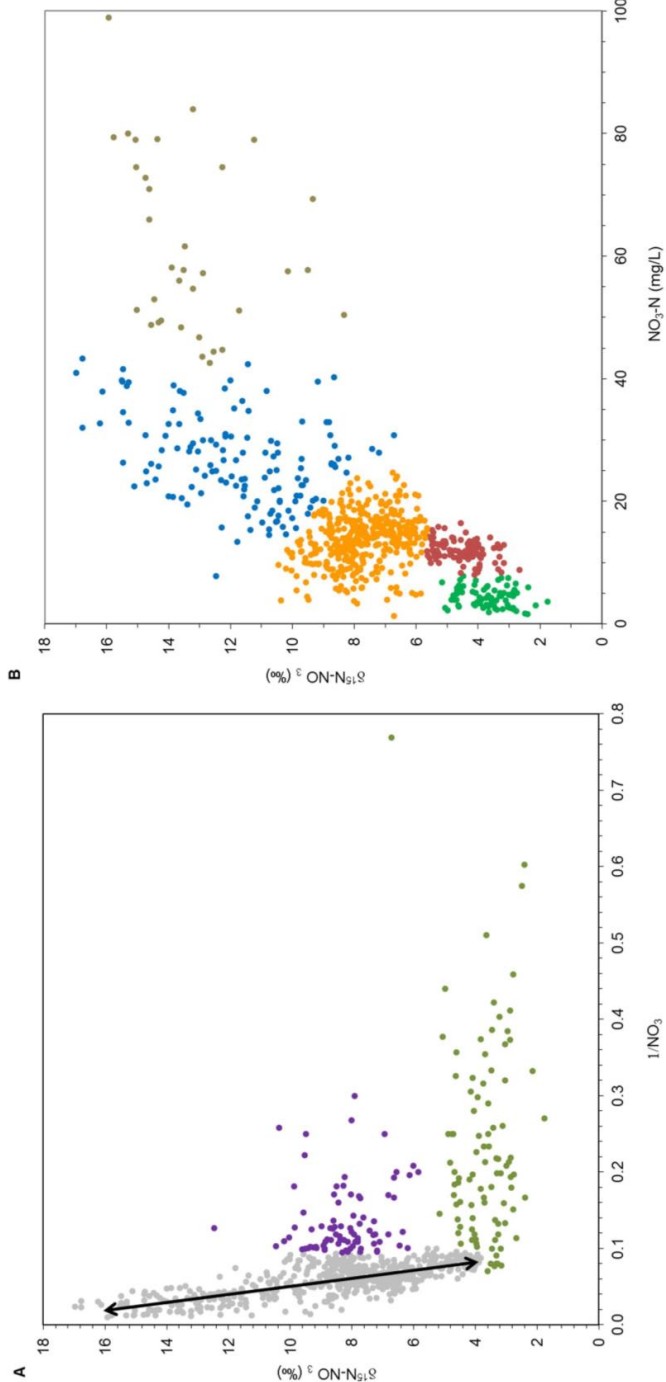





**Figure 3**: Nitrate $\delta^{18}O$ vs $\delta^{15}N$ cross-plot. Distribution of 19 well sites grouped by $\delta^{15}N$ range and $\delta^{18}O$.
Group 1a: $\delta^{15}N$ range (3 to 8‰). Group 1b: $\delta^{15}N$ range (+2 to +16‰) $\delta^{18}O$ full range. Group 2: $\delta^{15}N$ range
(+6 to +10‰). Group 3: $\delta^{15}N$ range (+9 to +16‰).

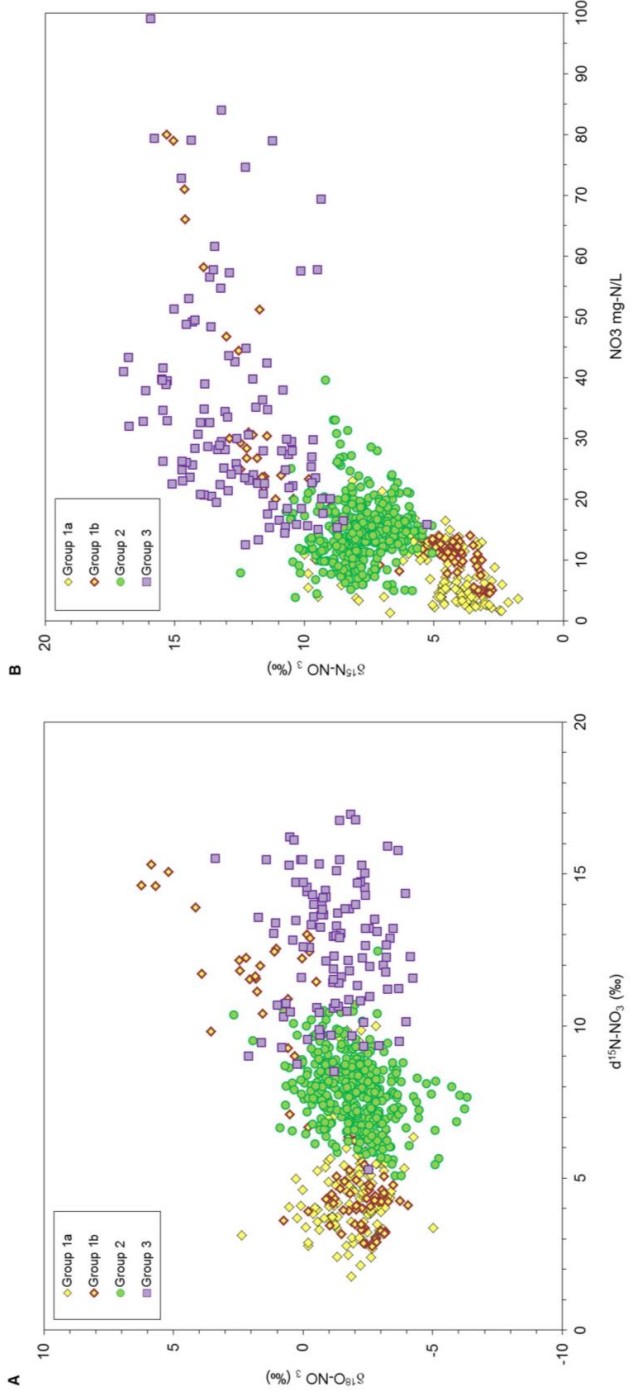






**Figure 4**: Spatial distribution of $\delta^{15}N$ source groupings, along with local agricultural land-use.








**Figure 5**: $\delta^{15}$N$_{-NO3}$ time series plots: A) No trend - stable (SD<±1.0), B) No trend - variable (SD>±1.0), C)
Enrichment trend, D) Depleting trend.





**Figure 6**: Spatial distribution of $\delta^{15}N$ trend groupings, along with agricultural land-use.





**Table 1**: Local synthetic fertilizer, poultry manure, soil N and leachate $\delta^{15}N$ values used in the Abbotsford area.

| Source | $\delta^{15}N$ (AIR) | Reference |
|---|---|---|
| Poultry Manure (total N) | +7.9 | Loo et al., 2017 |
| Poultry Manure (total N) | +8.1 | Wassenaar, 1995 |
| Poultry Manure (total N) | +7.9 | Wassenaar, 1995 |
| Urea (total N) | -0.7 | Loo et al., 2017 |
| NH4-NO3 (total N) | -2.8 | Loo et al., 2017 |
| NH4-SO4 (total N) | +0.3 | Loo et al., 2017 |
| Urea (total N) | -0.6 | Wassenaar, 1995 |
| NH4-SO4 (total N) | -0.9 | Wassenaar, 1995 |
| Soil N (total N) | +3.8 to +4.2 | Loo et al., 2017 |
| Soil N (total N) | +3.7 to +4.1 | Wassenaar, 1995 |
| Irrigation water - average (NO3-N) | +9.0 | Loo et al., 2017 |
| Weighted fertilizer treatment leachate (NO3-N) | +3.2±2.3 | Loo et al., 2017 |
| Weighted manure leachate (NO3-N) | +7.3±1.2 | Loo et al., 2017 |






**Table 2**. Bayesian clustering model of $NO_3^-$ and $\delta^{15}N$ means by class.

| Class | 1 | 2 | 3 | 4 | 5 |
|---|---|---|---|---|---|
| Mean ($NO_3^-$) | 4.4 | 13.2 | 13.5 | 22.9 | 55.2 |
| Mean ($\delta^{15}N$) | 3.7 | 5.4 | 7.9 | 10.7 | 13.2 |






**Table 3**: Nitrate isotopic Distribution and Trend grouping classification.

| | Source Grouping | | $\delta^{15}N$ Trend Grouping |
|---|---|---|---|
| 1a | $\delta^{15}N$ range (+3 to +8‰), $\delta^{18}O$ range (-5 to +2‰) | A | No trend - stable (SD< ±1.0 ‰) |
| 1b | $\delta^{15}N$ range (+2 to +16‰), $\delta^{18}O$ (-7 to +7‰) | B | No trend - variable (SD> ±1.0 ‰) |
| 2 | $\delta^{15}N$ range (+6 to +10‰), $\delta^{18}O$ range (-5 to +2‰) | C | Enrichment |
| 3 | $\delta^{15}N$ range (+9 to +16‰), $\delta^{18}O$ range (-5 to +2‰) | D | Depleting |




**Table 4**: Results summary with $^3$H/$^3$He groundwater ages in years (Wassenaar et al., 2006); Average water
column height (meters; mid-screen depth below average static water level); Isotopic Distribution and Trend
groupings; NO$_3$-N, $\delta^{15}$N and $\delta^{18}$O values (mean, standard deviation, and confidence intervals ($\alpha$=0.05)).

| Site ID | $^3$H/3He Age (yrs.) | Average Water Column | Source Group | Trend Group | NO$_3$-N‰ Mean | SD | CI | $\delta^{15}$N‰ Mean | SD | CI | $\delta^{18}$O‰ Mean | SD | CI |
|---|---|---|---|---|---|---|---|---|---|---|---|---|---|
| 91-03 | 3.5 | 2 | 2 | C | 17.2 | 3.8 | 1.0 | 7.0 | 0.6 | 0.2 | -2.0 | 1.0 | 0.3 |
| 91-07 | 2.7 | 1.8 | 1a | C | 13.2 | 3.3 | 0.9 | 5.4 | 1.5 | 0.5 | -2.6 | 0.8 | 0.3 |
| 91-10 | 3.2 | 3 | 3 | B | 33.2 | 11.6 | 3.0 | 13.1 | 1.6 | 0.5 | -0.9 | 1.4 | 0.4 |
| 91-15 | 5.94 | 7.2 | 2 | A | 12.1 | 3.7 | 1.0 | 8.9 | 1.0 | 0.3 | -1.4 | 0.8 | 0.3 |
| 94Q-14 | 4.2 | 6.3 | 2 | D | 7.7 | 1.9 | 0.5 | 8.0 | 0.8 | 0.3 | -0.9 | 0.8 | 0.3 |
| ABB-02 | 5.5 | 5 | 2 | A | 14.0 | 3.4 | 0.9 | 8.0 | 0.5 | 0.2 | -3.5 | 1.4 | 0.4 |
| ABB-03 | 0.9 | 5.2 | 2 | D | 12.4 | 3.9 | 1.0 | 7.5 | 1.2 | 0.4 | -1.3 | 1.6 | 0.5 |
| ABB-05 | 4.3 | 6.7 | 2 | A | 16.2 | 2.3 | 0.6 | 6.4 | 0.6 | 0.2 | -2.6 | 0.9 | 0.3 |
| FT5-12 | N/A | 2 | 2 | A | 16.1 | 6.4 | 1.7 | 8.4 | 0.9 | 0.3 | -1.9 | 1.0 | 0.3 |
| FT5-25 | N/A | 5.6 | 2 | C | 13.1 | 2.6 | 0.7 | 8.8 | 0.9 | 0.3 | -1.5 | 1.0 | 0.3 |
| PA-25 | 4.2 | 2.9 | 1a | B | 5.8 | 3.4 | 0.9 | 4.1 | 1.8 | 0.6 | -1.7 | 1.1 | 0.4 |
| PA-35 | 4.7 | 6.7 | 1a | B | 4.6 | 2.2 | 0.6 | 5.1 | 2.2 | 0.7 | -1.4 | 0.8 | 0.3 |
| PB-20 | 1.3 | 2.4 | 2 | B | 18.9 | 5.1 | 1.3 | 8.0 | 1.1 | 0.4 | -2.1 | 1.2 | 0.4 |
| PB-35 | 4.8 | 6.7 | 2 | D | 17.0 | 3.7 | 1.0 | 7.4 | 0.7 | 0.2 | -2.1 | 0.9 | 0.3 |
| PC-25 | 1.5 | 2.2 | 1b | C | 17.5 | 19.6 | 5.1 | 6.8 | 4.5 | 1.4 | -0.3 | 3.0 | 0.9 |
| PC-35 | 4.4 | 6.3 | 1b | C | 14.9 | 7.0 | 1.8 | 6.6 | 3.3 | 1.0 | -1.4 | 1.6 | 0.5 |
| US-02 | 1 | 4.6 | 3 | B | 38.6 | 22.8 | 6.0 | 11.4 | 2.3 | 0.7 | -1.7 | 1.7 | 0.5 |
| US-04 | 5 | 6.9 | 1a | C | 13.2 | 2.0 | 0.5 | 5.4 | 0.5 | 0.2 | -2.2 | 0.7 | 0.2 |
| US-05 | <1 | 1 | 3 | B | 27.3 | 10.1 | 3.2 | 13.4 | 2.2 | 0.8 | -1.3 | 1.2 | 0.4 |
