# Peer review of "High-frequency $NO_3^-$ isotope ( $\delta^{15}N$ , $\delta^{18}O$ ) patterns in groundwater recharge reveal that short-term changes in land use and precipitation influence nitrate contamination trends"

_Hydrology and Earth System Sciences, 2018_

## Referee Comment (RC1) · Anonymous Referee #1 · 18 Mar 2018

Review of:

High-frequency NO3- isotope ($\delta$15N, $\delta$18O) patterns in groundwater recharge reveal that short-term land use and climatic changes influence nitrate contamination trends Submitted to HESS (hess-2018-35) by Martin Suchy, Leonard I. Wassenaar, Gwyn Graham, and Bernie Zebarth

The submitted manuscript focusses on nitrate concentrations and there relation to N inputs from different natural and anthropogenic sources in the transboundary Abbotsford-

[Figure]

Sumas aquifer (ASA). Over a five-year period, they took monthly samples from newly-recharged groundwater for isotope and concentration analysis, to get insights into spatial and temporal nitrate dynamics.

Major comments:

Due to its clear structure and to the comprehensible English, the author's thoughts are easy to follow and the manuscript is good to read. The presentation of data in figures and tables could be improved (remarks below-mentioned).

The authors explained clearly the gaps of knowledge and there objectives in the introduction section. The used data including a five-year monitoring of nitrate concentrations (plus chloride as conservative tracer) and corresponding isotopic signatures for 19 monitoring wells presents an interesting dataset for a region which is partly impacted by high agricultural landuse and poultry.

Possibly, it would be useful to generate a depth profile which represents the specific sampling depth for each monitoring well. Thereby, different nitrate sources which appear relatively close (Figure 4, 91-03 und 91-07) could perhaps better explained.

Does exist a direct connection between dominant landuse type (blueberry, raspberry, poultry etc.) in the surrounding of the monitoring well and major nitrate source or rather microbial process dynamics? Please try to explain in more detail dominate N-sources for wells with a small distance but different $\delta$15N source grouping.

To get insights into residence time and the connection to surface water, it would have been nice to additionally measure deuterium and oxygen isotopes in water. Sad, that these additional variables are not part of this monitoring concept.

The manuscript represents an interesting five-year dataset which interpret the major nitrate sources on a spatial and temporal scale for the ASA which is highly impacted by cultivation and poultry input. The interpretation of the results is precise and clear. Some figures need further improvement. Despite the above mentioned questions and

comments, I'd suggest accepting the manuscript for publication in the Journal HESS if the authors are willing to address those questions and to apply major revisions.

Minor comments:

In chapter 3.1, the authors describe the dissolved oxygen content (DO – used abbreviation not explained) which is usually higher than 3 mg L-1. Unfortunately, I can't find further information in the Supplementary Table which is referred to.

Chapter 2.2: "The analytical uncertainty for $\delta$8O was $\pm 1.0$‰' – correct $\delta$18O

Chapter 3.2: first line "+7.9 $\pm$ .11‰' – correct/complete the last number (0.11‰

Chapter 3.2: What is the "Bayesian VVV"? Please briefly explain.

Figure 2A: It would be useful to see typical $\delta$15N signatures for dominate nitrate sources (endmember) on the left site of the diagram. Is it possible to add arrows with the typical range for manure, soil-N and synthetic fertilizer?

Figure 3: Correct the axes labels (delta, shift and units in breaks)

Figure 4: ECCC sites are presented in the legend but I can't find one single red dot in the scheme. Consider the uniform use of capitalization.

Figures generally: What is the reason of the used nomenclature from the monitoring wells? Is it necessary to use these abbreviations?

---

## Referee Comment (RC2) · Anonymous Referee #2 · 20 Apr 2018

General comments:

Nitrate contamination in groundwater is a widespread problem often associated with industrial agriculture. Many attempts to address excessive nitrate concentrations in groundwater by landuse management changes have yielded only sluggish or negligible success, indicating that our knowledge about sources and processes affecting nitrate in groundwater and the associated transit times are still rather incomplete.

The manuscript by Suchy et al. makes a highly valuable contribution to close this knowledge gap by providing excellent new insights into sources and processes affecting nitrate concentrations in young groundwater in the transboundary Abbotsford-Sumas aquifer. The determination of 700 nitrate isotope compositions for age-controlled groundwater (< 5 years old) collected between summer 2008 and spring 2013 yielded novel insights about sources of groundwater nitrate in a study area where the predominant nitrogen inputs have recently shifted somewhat from manure towards synthetic fertilizers. In addition, the authors were able to determine the effects of local crop rotations and disturbances due to their spatially and temporally intensive sampling strategy. Since nitrate contamination in this aquifer has previously been reported by Wassenaar (2005) and Wassenaar et al. (2006), the authors were also able to report on subtle shits of nitrate sources on decadal time scales. These new findings make a highly valuable contribution to enhancing the understanding of sources, processes, and time-lines of nitrate contamination of groundwater and hence will be of high interest to the readership of Hydrology and Earth System Sciences.

The current draft manuscript contains several moderate and numerous minor deficiencies that should be addressed prior to acceptance of this manuscript, including the following:

- In the introduction, the authors outline the differences in ïĄď15N values between synthetic fertilizers and manure-derived nitrate and also elaborate on the oxygen isotope ratios on synthetic nitrate-containing fertilizers. What is missing is a short description of oxygen isotope ratios of nitrate expected from nitrification of organic N, urea, and ammonium-sulfate in dependence of the ïĄď18O value of local water in the unsaturated and saturated zones. It is important to add this information to the introduction to provide the readership with a full background on the usefulness of isotopic tracers for distinguishing sources and processes affecting nitrate in the study area.

- Due to the importance of landuse changes and the trends away from manure additions towards synthetic fertilizers, is appears highly desirable to describe the changes in agricultural practices at the study site in a bit more detail in this manuscript.

-The authors have made an excellent effort to constrain their sampling of the aquifer to wells that access aerobic groundwater of average age of less than 5 years to link the detected trends to recent agricultural activities. While this argumentation holds most likely true for the water-saturated portion of the study area, it is important to realize that a similar reasoning is not entirely valid for the unsaturated zone including the soils. In the water-unsaturated and soil zones, "subsurface biogeochemical processes" are certainly ongoing with N immobilization and re-mineralization potentially delaying N transfers for years or decades (see for instance Sebilo et al. (2013): Long-term fate of nitrate fertilizer in agricultural soils; PNAS 110(45): 18185-18189), although the manuscript text on line 134 seems to suggest the opposite. Throughout the manuscript, the authors should make it more clear that their approach provides only very limited insights into N cycling and its transit times in the soil and water-unsaturated zones.

- In Figure 2a and associated text on lines 216-224, the authors assign the nitrate isotope data to three nitrate sources. Nitrate in irrigation water (ïĄď'15N of +9 ‰ and manure (ïĄď'15N of +8 ‰ are the sources with the highest ïĄď'15N values, but Figure 2a shows numerous samples with ïĄď'15N values between 10 and 17 ‰Å short explanation for these elevated ïĄď'15N values is desirable at this point in the manuscript.

- Line 241: The mean ïĄď'15N value of +5.0 ‰ is not very close to that of synthetic fertilizers (ïĄď'15N near 0 ‰. Is it possible that intensive N cycling in the soil with associated N isotope effects causes a shift to higher ïĄď'15N values in the seepage water nitrate? If this is a requirement to explain the data patterns, this should be acknowledged in the text of this manuscript.

- In my view, the evidence for climatic impacts on trends in the chemical and isotopic composition of groundwater nitrate presented in this manuscript is very weak (e.g. lines 323-325) and is mainly based on references to data presented elsewhere rather than in this manuscript. I am not convinced that a few years on increased precipitation (2008-2011) justify mentioning "climatic changes" in the title of this manuscript especially since no climate data are presented.

[Figure]

- Table 1 lists another nitrate source, namely nitrate-containing irrigation water with a ïĄď15N of +9 ‰ evidently derived from manure-applications. Throughout the text, this nitrate source receives very little attention. Is it not relevant?

In addition there are a number of minor deficiencies that include the following more specific comments:

- Line 47-50: It should be made clear that atmospheric nitrate inputs are not leached into the groundwater conservatively, but usually undergo intensive recycling via immobilization and ammonification + nitrification in the unsaturated zone prior to reaching the groundwater zone.

- In line 58, the authors state that manure-derived nitrate has ïĄď15N typically >10‰ but subsequently report on line 61 that the ïĄď15N of poultry manure in the study area is closer to 8 ‰. What explains the discrepancy? Is the former range mainly for cattle manure?

- In lines 101-107 the aquifer is well described, but one essential piece of information, the depth of the water table below ground surface is not clearly revealed. The authors should add this information in a more transparent fashion;

- To support the statement that the aquifer is largely under aerobic conditions it would be beneficial to add dissolved oxygen concentrations to the manuscript (for instance in table 4).

- In section 2.2, it would be useful to list the depths of water table below ground surface for the 19 selected monitoring wells.

- Lines 146-7: The measurement uncertainties for concentration analyses (e.g. nitrate, chloride) should be provided;

- Section 3.1: throughout this section it would be more correct to speak about nitrate concentrations of groundwater obtained from wells (since wells have no nitrate concentrations);

- Line 178: state by how much the nitrate concentration increased over the 5-year observation period; if you exclude the three wells mentioned on line 184, is there still an increase in nitrate concentrations for the groundwater from the remaining 16 wells?

-Line 215: The end-member with low ïĄ𝑑'15N values appears to have "intermediate" nitrate concentrations;

- Lines 221-222: The rational why the Bayesian clustering model that suggest 5 group-ings results in 4 distinct groups (line 223) is not clear to me.

- Line 236-239: Are these 4 groups shown in any Figures? Also, to which category belong the samples with ïĄ𝑑'15N values between 10 and 17 ‰

- Line 278: I suspect not the soil N is flushed to the aquifer, but nitrate derived from nitrification of soil N. - Line 288: Can you quantify the extent of this decrease in ïĄ𝑑'15N over 5 years? How does it compare to the long-term decrease in groundwater nitrate ïĄ𝑑'15N observed since 1995?

- Line 328: Logic unclear: if 14-N was preferentially volatilized, should the remaining N compound not be enriched in 15-N?

- Line 359: Please quantify the extent of the observed decrease in ïĄ𝑑'15N values.

The manuscript is written in excellent English, it follows a logical sequence and is hence very well organized, and the objectives are clearly stated. The applied meth-ods are leading-edge and are sufficiently described. Previous literature is exhaustively considered. Figures and tables are of good quality with minor deficiencies listed be-low. Hence, if the authors are able to address the limitations identified in this review, publication of this manuscript after moderate revisions is recommended.

Additional technical comments:

Line Comment

I suspect you did not measure "recharge" directly, but shallow groundwater up to 5

years old; please re-word accordingly; one or two more recent references that are less than 10 years old may be desirable; do you mean "nitrate" isotopes or also other isotopic parameters? If the latter, please mention which other isotopes?

are these ïĄd'15N values representative for this study site?

56, 71, 77, 110 et al.

& following I suspect the numbers in brackets are N-P-K values for synthetic fertilizers, but this may need to be explained to the readership.

add a reference to support this statement; it would be advantageous to spell out the fertilizer sources used in the study area; a more detailed explanation on how the oxygen isotope ratios of nitrate derived from nitrification are controlled is needed here; samplings (should likely be plural)

add a reference for "winter-biased recharge";

. . . with a focus on "shallow groundwater" from water table wells . . .

something seems wrong or duplicated here: ..." isotope nitrate and isotope . . ."; also "processes" should be plural;

"unpublished data" should be moved inside the brackets; delete "surface"

. . . of nitrate "in groundwater" . . .

if possible add average depths for deep wells and average nitrate-N concentrations; not the wells are aerobic, but the groundwater obtained from the wells; in ïАď18O the "1" appears to be missing; does this refer to "nitrate concentrations"?

indicate in which months the first major recharge occurs? Is it late fall?

are vadose zone infiltration lag-times similar for all sites?

the groundwater is aerobic, not the wells; I did not find a supplementary table; delete "a"

what does this limited variability indicate? Longer transit times through the unsaturated zone?

throughout this section, the ïАď15N values are for nitrate in groundwater, not for wells.

210, 213, 214 no need to report data with 2 decimal places given the measurement uncertainty of this parameter; what is meant with "like group 1a values". ïАď15N of 6.7 ‰ is not like 5.0 ‰ and even further away from synthetic fertilizer ïАď15N values of 0 ‰

groundwater flow paths are neither shown in Figure 1 nor in 4;

. . . influenced "the nitrate contamination level" in these wells; replace "isotopically" with 15N-enriched; could you not have microbial transformations but with negligible isotope fractionation? Almost all transformations in the N cycle are microbially mediated;

I could not find the supplementary table; it is not possible to enrich a delta value. Also, do you mean enrichment in 14-N or

15-N?

increasing trend for which parameter?

Wasenaar et al. (2006)

ïĄď18O of nitrate were anaerobic conditions detected based on DO concentrations? If so what were the DO concentration ranges?

and Wassenaar et al. (2006)

depletion of 15-N in what: in groundwater nitrate? Why 15-N depletion if you previously talked about denitrification?

363-4 this is new information that was not previously provided in the Results & Discussion section.

do you men enrichment in 14-N or 15-N of nitrate?

do you mean concentrations and isotopic compositions of nitrate?

do you mean "groundwater" nitrate?

the inset map requires a distance bar (in km); units are missing for ïĄď15N

Table 2: why are concentrations listed here as nitrate, when throughout the rest of the manuscript they are given as nitrate-N? Also units are missing.

Depletion (rather than depleting)?

nitrate concentration unit is wrong: mg/L rather than ‰

Please also note the supplement to this comment:
https://www.hydrol-earth-syst-sci-discuss.net/hess-2018-35/hess-2018-35-RC2- supplement.pdf

---

## Author Comment (AC1) · 22 May 2018

Referee #1

Major comments:

Due to its clear structure and to the comprehensible English, the author's thoughts are easy to follow and the manuscript is good to read. The presentation of data in figures and tables could be improved (remarks below-mentioned). The authors explained clearly the gaps of knowledge and their objectives in the introduction section. The used

data including a five-year monitoring of nitrate concentrations (plus chloride as conservative tracer) and corresponding isotopic signatures for 19 monitoring wells presents an interesting dataset for a region which is partly impacted by high agricultural land use and poultry.

Possibly, it would be useful to generate a depth profile which represents the specific sampling depth for each monitoring well. Thereby, different nitrate sources which appear relatively close (Figure 4, 91-03 und 91-07) could perhaps better explained. RESPONSE: Table 4 shows the average water column height in meters as defined at the mid-screen depth below average static water, which is more relevant to the 3H-3He groundwater 'ages' than a depth profile would provide, due to large variations in unsaturated zone thickness across the aquifer. We elaborate our reference the water column height values in Table 4.

Does exist a direct connection between dominant landuse type (blueberry, raspberry, poultry etc.) in the surrounding of the monitoring well and major nitrate source or rather microbial process dynamics? Please try to explain in more detail dominate N-sources for wells with a small distance but different d15N source grouping. RESPONSE: Field specific nutrient management practices and land use are more dominant than microbial drivers in this highly aerobic aquifer (assuming the reviewer is hinting at potential anaerobic denitrification). Wells are mostly located adjacent to fields with different land use practices as depicted in Fig 4, so the groundwater flow direction is also relevant. We already strongly emphasize the close connection of nitrate concentration in recharge and its isotopic values to changeable near-field land use practices.

To get insights into residence time and the connection to surface water, it would have been nice to additionally measure deuterium and oxygen isotopes in water. Sad, that these additional variables are not part of this monitoring concept. RESPONSE: Previous publications (i.e. Wassenaar 1995) reported extensive d18O and d2H for groundwater across the aquifer at all depths that show characteristic fall and winter wet season dominated recharge and very little isotopic variance ($\sim$+/-0.4 permil for 18O). The

stable isotopes reveal the relative importance of seasonality of recharge compared to precipitation, but do not provide any control on groundwater residence time like 3H-3He does. Of course, it would have been nice to include many more geochemical covariates, but our budget was limited and focused on primary aims.

The manuscript represents an interesting five-year dataset which interpret the major nitrate sources on a spatial and temporal scale for the ASA which is highly impacted by cultivation and poultry input. The interpretation of the results is precise and clear.

Some figures need further improvement. Despite the above-mentioned questions and comments, I'd suggest accepting the manuscript for publication in the Journal HESS if the authors are willing to address those questions and to apply major revisions.

Minor comments:

In chapter 3.1, the authors describe the dissolved oxygen content (DO – used abbreviation not explained) which is usually higher than 3 mg L-1. Unfortunately, I can't find further information in the Supplementary Table which is referred to. RESPONSE: Agree. We added dissolved oxygen concentrations for each sampling to the Supplemental Table.

Chapter 2.2: "The analytical uncertainty for $\_8O$ was $\pm 1.0‰\$'' - -correct\_18O RESPONSE : Corrected.$

Chapter 3.2: first line "+7.9 $\pm .11‰\$'' -- correct/complete the last number (0.11\ RESPONSE : Corrected.$

Chapter 3.2: What is the "Bayesian VVV"? Please briefly explain. RESPONSE: Agree. Modified the text to clarify.

Figure 2A: It would be useful to see typical d15N signatures for dominate nitrate sources (endmember) on the left site of the diagram. Is it possible to add arrows with the typical range for manure, soil-N and synthetic fertilizer? RESPONSE: Figure modified.

Figure 3: Correct the axes labels (delta, shift and units in breaks) RESPONSE: Corrected.

Figure 4: ECCC sites are presented in the legend but I can't find one single red dot in the scheme. Consider the uniform use of capitalization. RESPONSE: Agreed. We removed the red dots in Figure 4.

Figures generally: What is the reason of the used nomenclature from the monitoring wells? Is it necessary to use these abbreviations? RESPONSE: Disagree. All previous published studies used these well identifiers. Re-labelling well names will cause confusion going forward by requiring an additional 'key' to reference previously published datasets.

---

## Author Comment (AC2) · 22 May 2018

Referee #2

Nitrate contamination in groundwater is a widespread problem often associated with industrial agriculture. Many attempts to address excessive nitrate concentrations in groundwater by land use management changes have yielded only sluggish or negligible success, indicating that our knowledge about sources and processes affecting nitrate in groundwater and the associated transit times are still rather incomplete.

The manuscript by Suchy et al. makes a highly valuable contribution to close this knowledge gap by providing excellent new insights into sources and processes affecting nitrate concentrations in young groundwater in the transboundary Abbotsford-Sumas aquifer. The determination of 700 nitrate isotope compositions for age-controlled groundwater (< 5 years old) collected between summer 2008 and spring 2013 yielded novel insights about sources of groundwater nitrate in a study area where the predominant nitrogen inputs have recently shifted somewhat from manure towards synthetic fertilizers. In addition, the authors were able to determine the effects of local crop rotations and disturbances due to their spatially and temporally intensive sampling strategy.

Since nitrate contamination in this aquifer has previously been reported by Wassenaar (2005) and Wassenaar et al. (2006), the authors were also able to report on subtle shits of nitrate sources on decadal time scales. These new findings make a highly valuable contribution to enhancing the understanding of sources, processes, and timelines of nitrate contamination of groundwater and hence will be of high interest to the readership of Hydrology and Earth System Sciences.

The current draft manuscript contains several moderate and numerous minor deficiencies that should be addressed prior to acceptance of this manuscript, including the following:

In the introduction, the authors outline the differences in d15N values between synthetic fertilizers and manure-derived nitrate and also elaborate on the oxygen isotope ratios on synthetic nitrate-containing fertilizers. What is missing is a short description of oxygen isotope ratios of nitrate expected from nitrification of organic N, urea, and ammonium-sulfate in dependence of the d18O value of local water in the unsaturated and saturated zones. It is important to add this information to the introduction to provide the readership with a full background on the usefulness of isotopic tracers for distinguishing sources and processes affecting nitrate in the study area. RESPONSE: Agreed. Added sentences to introduction on the expected d18O from nitrification of organic N sources based on this aquifers' isotope water isotope data.

Due to the importance of land use changes and the trends away from manure additions towards synthetic fertilizers, is appears highly desirable to describe the changes in agricultural practices at the study site in a bit more detail in this manuscript. RESPONSE: Agreed. We made changes to the introduction (added Lines 122-125) describing the longer-term general shifts in land use practices.

The authors have made an excellent effort to constrain their sampling of the aquifer to wells that access aerobic groundwater of average age of less than 5 years to link the detected trends to recent agricultural activities. While this argumentation holds most likely true for the water-saturated portion of the study area, it is important to realize that a similar reasoning is not entirely valid for the unsaturated zone including the soils. In the water-unsaturated and soil zones, "subsurface biogeochemical processes" are certainly ongoing with N immobilization and re-mineralization potentially delaying N transfers for years or decades (see for instance Sebilo et al. (2013): Long-term fate of nitrate fertilizer in agricultural soils; PNAS 110(45): 18185-18189), although the manuscript text on line 134 seems to suggest the opposite. Throughout the manuscript, the authors should make it more clear that their approach provides only very limited insights into N cycling and its transit times in the soil and water-unsaturated zones. RESPONSE: While we agree with this comment as a general observation, we must respectfully counter that unsaturated zone biogeochemical reprocessing of N sources is not necessarily the case and of a significant magnitude everywhere. As described here and in previous work, this aquifer has thin and poorly developed organic soil and is comprised of coarse sand, gravel and cobbles with poor water retention no capacity to develop anaerobic zone potentials for denitrification. This is evident by fully aerobic conditions nearly everywhere in the aquifer. Moreover, as demonstrated in detailed unsaturated zone isotope tracer experimental work done over this aquifer by Loo et al, they showed that the N isotopic fingerprints of manure versus fertilizer applications were retained in the unsaturated zone, albeit with dampened signals due to N reservoir mixing. If anything, this case study refutes the general idea that source N signals are always be masked by unsaturated zone biogeochemical processes. N

cycling processes, or lack thereof, are site specific. However, as noted any shifts in sources could indeed take a longer time to be seen due to source changes and vadose zone mixing, which is what we describe (subtle decadal shifts in sources).

In Figure 2a and associated text on lines 216-224, the authors assign the nitrate isotope data to three nitrate sources. Nitrate in irrigation water (d15N of +9 ‰ and manure (d15N of +8 ‰ are the sources with the highest 15N values, but Figure 2a shows numerous samples with d15N values between 10 and 17 ‰À short explanation for these elevated d15N values is desirable at this point in the manuscript. RESPONSE: We added a sentence to refer to Wassenaar (1995) that showed d15N values up to +14 ‰ in nitrate affected by ammonia volatilization and/or nitrification around poultry storage and spreading operations, despite rather limited data. Higher localized d15N values for nitrate derived from poultry manure are likely, but not the manure solids themselves.

Line 241: The mean d15N value of +5.0 ‰ is not very close to that of synthetic fertilizers (d15N near 0 ‰Is it possible that intensive N cycling in the soil with associated N isotope effects causes a shift to higher d15N values in the seepage water nitrate? If this is a requirement to explain the data patterns, this should be acknowledged in the text of this manuscript. RESPONSE: Disagree. We noted that the fertilizers used in the area have a mean d15N of +3.2+/-2.3 as reported in Loo et al (2017). This is consistent with our interpretation.

In my view, the evidence for climatic impacts on trends in the chemical and isotopic composition of groundwater nitrate presented in this manuscript is very weak (e.g. lines 323-325) and is mainly based on references to data presented elsewhere rather than in this manuscript. I am not convinced that a few years on increased precipitation (2008-2011) justify mentioning "climatic changes" in the title of this manuscript especially since no climate data are presented. RESPONSE: Agreed. We replaced climatic changes with 'precipitation changes'. See the citation given for details.

[Figure]

Table 1 lists another nitrate source, namely nitrate-containing irrigation water with a ïAËŽd'15N of +9 ‰ evidently derived from manure-applications. Throughout the text, this nitrate source receives very little attention. Is it not relevant? RESPONSE: True. But as noted in previous works on the aquifer, groundwater irrigation water (i.e. 15N enriched nitrate laden groundwater) is only applied during Jun-Aug during peak ET and maximum crop nutrient uptake, resulting in limited, summer recharge by irrigation water. The isotope signal is anyhow not readily distinguishable from contemporary manure sources.

In addition, there are a number of minor deficiencies that include the following more specific comments:

Line 47-50: It should be made clear that atmospheric nitrate inputs are not leached into the groundwater conservatively, but usually undergo intensive recycling via immobilization and ammonification + nitrification in the unsaturated zone prior to reaching the groundwater zone. RESPONSE: See above response to vadose zone N recycling comments.

In line 58, the authors state that manure-derived nitrate has d15N typically >10‰ but subsequently report on line 61 that the d15N of poultry manure in the study area is closer to 8 ‰Ẇhat explains the discrepancy? Is the former range mainly for cattle manure? RESPONSE: See above response to Reviewer #1 – we added a sentence about higher d15N values encountered up to +14 per mil observed in nitrates formed around poultry operations.

In lines 101-107 the aquifer is well described, but one essential piece of information, the depth of the water table below ground surface is not clearly revealed. The authors should add this information in a more transparent fashion; RESPONSE: See above response to Reviewer #1. Water table depth is not a static value. It is highly variable spatially and over time. Instead, Table 4 shows the average water column height in meters as defined at the mid-screen depth below average static water, which is more

relevant to the 3H-3He groundwater 'ages' than a depth profile would provide due to large variations in unsaturated zone depth across the aquifer. We believe this is a better way to show this information, and prefer to keep it.

To support the statement that the aquifer is largely under aerobic conditions it would be beneficial to add dissolved oxygen concentrations to the manuscript (for instance in table 4). RESPONSE: Agreed – also noted by Reviewer #1 – dissolved O2 was added to SM Table.

In section 2.2, it would be useful to list the depths of water table below ground surface for the 19 selected monitoring wells. RESPONSE: See above, same comment.

Lines 146-7: The measurement uncertainties for concentration analyses (e.g. nitrate, chloride) should be provided; RESPONSE: Agreed. Added the MDL.

Section 3.1: throughout this section it would be more correct to speak about nitrate concentrations of groundwater obtained from wells (since wells have no nitrate concentrations); RESPONSE: Agreed. Changed where appropriate.

Line 178: state by how much the nitrate concentration increased over the 5-year observation period; if you exclude the three wells mentioned on line 184, is there still an increase in nitrate concentrations for the groundwater from the remaining 16 wells? RESPONSE: Addressed.

Line 215: The end-member with low d15N values appears to have "intermediate" nitrate concentrations; RESPONSE: Agreed, changed as appropriate.

Lines 221-222: The rational why the Bayesian clustering model that suggest 5 groupings results in 4 distinct groups (line 223) is not clear to me. RESPONSE: Addressed. The first two Bayesian groups were amalgamated as they suggested the same isotopic source, but with partial enrichment.

Line 236-239: Are these 4 groups shown in any Figures? Also, to which category belong the samples with d15N values between 10 and 17 ‰ RESPONSE: Addressed.

Figure 3 and d15N 9 to 16 is Group 3.

Line 278: I suspect not the soil N is flushed to the aquifer, but nitrate derived from nitrification of soil N. – RESPONSE: See response to vadose zone comment above.

Line 288: Can you quantify the extent of this decrease in d15N over 5 years? How does it compare to the long-term decrease in groundwater nitrate d15N observed since 1995? RESPONSE: Addressed.

Line 328: Logic unclear: if 14N was preferentially volatilized, should the remaining N compound not be enriched in 15N? RESPONSE: Fixed.

Line 359: Please quantify the extent of the observed decrease in d15N values. RE-SPONSE: Addressed. The manuscript is written in excellent English, it follows a logical sequence and is hence very well organized, and the objectives are clearly stated. The applied methods are leading-edge and are sufficiently described. Previous literature is exhaustively considered.

Figures and tables are of good quality with minor deficiencies listed below. Hence, if the authors are able to address the limitations identified in this review, publication of this manuscript after moderate revisions is recommended.

Additional technical comments:

Line Comment

20 I suspect you did not measure "recharge" directly, but shallow groundwater up to 5 years old; please re-word accordingly; RESPONSE: Agreed. Changed.

35 one or two more recent references that are less than 10 years old may be desirable; RESPONSE: Agreed. Newer citations added.

43 do you mean "nitrate" isotopes or also other isotopic parameters? If the latter, please mention which other isotopes? RESPONSE: Nitrate. Re-worded in the paragraph.

[Figure]

53 are these d15N values representative for this study site? RESPONSE: Yes. As cited in Loo et al.

62 & following I suspect the numbers in brackets are N-P-K values for synthetic fertilizers, but this may need to be explained to the readership. RESPONSE: Corrected.

65 add a reference to support this statement; RESPONSE: Reference added.

67 it would be advantageous to spell out the fertilizer sources used in the study area; RESPONSE: Corrected.

70 a more detailed explanation on how the oxygen isotope ratios of nitrate derived from nitrification are controlled is needed here; RESPONSE: Added sentence to clarify.

75 samplings (should likely be plural) RESPONSE: Corrected.

80 add a reference for "winter-biased recharge"; RESPONSE: Corrected.

84 . . . with a focus on "shallow groundwater" from water table wells . . . RESPONSE: Corrected.

86 something seems wrong or duplicated here: ..." isotope nitrate and isotope . . ."; also "processes" should be plural; RESPONSE: Corrected.

100 "unpublished data" should be moved inside the brackets; RESPONSE: Corrected.

105 delete "surface" RESPONSE: Corrected.

119 . . . of nitrate "in groundwater" . . . RESPONSE: Corrected.

121 if possible add average depths for deep wells and average nitrate-N concentrations; RESPONSE: Corrected.

132 not the wells are aerobic, but the groundwater obtained from the wells; RESPONSE: Corrected.

157 in d18O the "1" appears to be missing; RESPONSE: It is there.

168 does this refer to "nitrate concentrations"? RESPONSE: Corrected.

193 indicate in which months the first major recharge occurs? Is it late fall? RESPONSE: Clarified.

197 are vadose zone infiltration lag-times similar for all sites? RESPONSE: Corrected.

199 the groundwater is aerobic, not the wells; I did not find a supplementary table; RESPONSE: Wording corrected and added mean DO concentration. Added DO to the SM table.

200 delete "a" RESPONSE: Corrected.

206 what does this limited variability indicate? Longer transit times through the unsaturated zone? RESPONSE: addressed.

208 throughout this section, the d15N values are for nitrate in groundwater, not for wells. RESPONSE: Corrected.

210, 213, 214 no need to report data with 2 decimal places given the measurement uncertainty of this parameter; RESPONSE: Corrected.

250 what is meant with "like group 1a values". d15N of 6.7 ‰ is not like 5.0 ‰ and even further away from synthetic fertilizer d15N values of 0 ‰ RESPONSE: Corrected.

261 groundwater flow paths are neither shown in Figure 1 nor in 4; RESPONSE: Corrected.

270 . . . influenced "the nitrate contamination level" in these wells; RESPONSE: Corrected.

278 replace "isotopically" with 15N-enriched; RESPONSE: Corrected.

297 could you not have microbial transformations but with negligible isotope fractionation?Almost all transformations in the N cycle are microbially mediated; RESPONSE: Yes but with continuous input of groundwater N, resulting in no appreciable buildup of

enriched N. But our N inputs are seasonal, driven by winter recharge.

305 I could not find the supplementary table; RESPONSE: It is there.

314 it is not possible to enrich a delta value. Also, do you mean enrichment in 14-N or 15-N? RESPONSE: Corrected

323 increasing trend for which parameter? RESPONSE: 15N. Corrected

334 Wassenaar et al. (2006) RESPONSE: Corrected.

336 d18O of nitrate RESPONSE: Corrected.

338 were anaerobic conditions detected based on DO concentrations? If so what were the DO concentration ranges? RESPONSE: Corrected and added to SM as per other reviewers.

342 and Wassenaar et al. (2006) RESPONSE: Corrected

343 depletion of 15N in what: in groundwater nitrate? Why 15-N depletion if you previously talked about denitrification? RESPONSE: Corrected.

363-4 this is new information that was not previously provided in the Results & Discussion section. RESPONSE: It is evident in the maps presented beforehand. Added text to refer to this map.

376 do you men enrichment in 14-N or 15-N of nitrate? RESPONSE: Fixed.

373 do you mean concentrations and isotopic compositions of nitrate? RESPONSE: Corrected

390 do you mean "groundwater" nitrate? RESPONSE: Corrected

518 the inset map requires a distance bar (in km); RESPONSE: Corrected

541 units are missing for d15N RESPONSE: Corrected

544 Table 2: why are concentrations listed here as nitrate, when throughout the rest

[Figure]

of the manuscript they are given as nitrate-N? Also units are missing. RESPONSE: Corrected

547 Depletion (rather than depleting)? RESPONSE: Corrected

550 nitrate concentration unit is wrong: mg/L rather than ‰ RESPONSE: Corrected

Please also note the supplement to this comment:
https://www.hydrol-earth-syst-sci-discuss.net/hess-2018-35/hess-2018-35-AC2-supplement.pdf
* * *
[Figure]

**Supplement:**

**Referee #1**

**Major comments:**

Due to its clear structure and to the comprehensible English, the author's thoughts are easy to follow and the manuscript is good to read. The presentation of data in figures and tables could be improved (remarks below-mentioned). The authors explained clearly the gaps of knowledge and their objectives in the introduction section. The used data including a five-year monitoring of nitrate concentrations (plus chloride as conservative tracer) and corresponding isotopic signatures for 19 monitoring wells presents an interesting dataset for a region which is partly impacted by high agricultural land use and poultry.

**Possibly, it would be useful to generate a depth profile which represents the specific sampling depth for each monitoring well. Thereby, different nitrate sources which appear relatively close (Figure 4, 91-03 und 91-07) could perhaps better explained.**
RESPONSE: Table 4 shows the average water column height in meters as defined at the mid-screen depth below average static water, which is more relevant to the $^3H$-$^3He$ groundwater 'ages' than a depth profile would provide, due to large variations in unsaturated zone thickness across the aquifer. We elaborate our reference the water column height values in Table 4.

**Does exist a direct connection between dominant landuse type (blueberry, raspberry, poultry etc.) in the surrounding of the monitoring well and major nitrate source or rather microbial process dynamics? Please try to explain in more detail dominate N-sources for wells with a small distance but different d15N source grouping.**
RESPONSE:  Field specific nutrient management practices and land use are more dominant than microbial drivers in this highly aerobic aquifer (assuming the reviewer is hinting at potential anaerobic denitrification). Wells are mostly located adjacent to fields with different land use practices as depicted in Fig 4, so the groundwater flow direction is also relevant. We already strongly emphasize the close connection of nitrate concentration in recharge and its isotopic values to changeable near-field land use practices.

**To get insights into residence time and the connection to surface water, it would have been nice to additionally measure deuterium and oxygen isotopes in water. Sad, that these additional variables are not part of this monitoring concept.**
RESPONSE:  Previous publications (i.e. Wassenaar 1995) reported extensive d18O and d2H for groundwater across the aquifer at all depths that show characteristic fall and winter wet season dominated recharge and very little isotopic variance (~+/-0.4 permil for 18O). The stable isotopes reveal the relative importance of seasonality of recharge compared to precipitation, but do not provide any control on groundwater residence time like $^3H$-$^3He$ does.  Of course, it would have been nice to include many more geochemical covariates, but our budget was limited and focused on primary aims.

The manuscript represents an interesting five-year dataset which interpret the major nitrate sources on a spatial and temporal scale for the ASA which is highly impacted by cultivation and poultry input. The interpretation of the results is precise and clear.

Some figures need further improvement. Despite the above-mentioned questions and comments, I'd suggest accepting the manuscript for publication in the Journal HESS if the authors are willing to address those questions and to apply major revisions.

Minor comments:

**In chapter 3.1, the authors describe the dissolved oxygen content (DO – used abbreviation not explained) which is usually higher than 3 mg L-1. Unfortunately, I can't find further information in the Supplementary Table which is referred to.**
**RESPONSE:** Agree. We added dissolved oxygen concentrations for each sampling to the Supplemental Table.

**Chapter 2.2: "The analytical uncertainty for _8O was ±1.0‰´' – correct _18O**
**RESPONSE:** Corrected.

**Chapter 3.2: first line "+7.9 ± .11‰´' – correct/complete the last number (0.11‰**
**RESPONSE:** Corrected.

**Chapter 3.2: What is the "Bayesian VVV"? Please briefly explain.**
**RESPONSE:** Agree. Modified the text to clarify.

**Figure 2A: It would be useful to see typical d15N signatures for dominate nitrate sources (endmember) on the left site of the diagram. Is it possible to add arrows with the typical range for manure, soil-N and synthetic fertilizer?**
**RESPONSE:** Figure modified.

**Figure 3: Correct the axes labels (delta, shift and units in breaks)**
**RESPONSE:** Corrected.

**Figure 4: ECCC sites are presented in the legend but I can't find one single red dot in the scheme. Consider the uniform use of capitalization.**
**RESPONSE:** Agreed. We removed the red dots in Figure 4.

**Figures generally: What is the reason of the used nomenclature from the monitoring wells? Is it necessary to use these abbreviations?**
**RESPONSE:** Disagree. All previous published studies used these well identifiers. Re-labelling well names will cause confusion going forward by requiring an additional 'key' to reference previously published datasets.

Nitrate contamination in groundwater is a widespread problem often associated with industrial agriculture. Many attempts to address excessive nitrate concentrations in groundwater by land use management changes have yielded only sluggish or negligible success, indicating that our knowledge about sources and processes affecting nitrate in groundwater and the associated transit times are still rather incomplete.

The manuscript by Suchy et al. makes a highly valuable contribution to close this knowledge gap by providing excellent new insights into sources and processes affecting nitrate concentrations in young groundwater in the transboundary Abbotsford-Sumas aquifer. The determination of 700 nitrate isotope compositions for age-controlled groundwater (< 5 years old) collected between summer 2008 and spring 2013 yielded novel insights about sources of groundwater nitrate in a study area where the predominant nitrogen inputs have recently shifted somewhat from manure towards synthetic fertilizers. In addition, the authors were able to determine the effects of local crop rotations and disturbances due to their spatially and temporally intensive sampling strategy.

Since nitrate contamination in this aquifer has previously been reported by Wassenaar (2005) and Wassenaar et al. (2006), the authors were also able to report on subtle shits of nitrate sources on decadal time scales. These new findings make a highly valuable contribution to enhancing the understanding of sources, processes, and timelines of nitrate contamination of groundwater and hence will be of high interest to the readership of Hydrology and Earth System Sciences.

The current draft manuscript contains several moderate and numerous minor deficiencies that should be addressed prior to acceptance of this manuscript, including the following:

**In the introduction, the authors outline the differences in d15N values between synthetic fertilizers and manure-derived nitrate and also elaborate on the oxygen isotope ratios on synthetic nitrate-containing fertilizers. What is missing is a short description of oxygen isotope ratios of nitrate expected from nitrification of organic N, urea, and ammonium-sulfate in dependence of the d18O value of local water in the unsaturated and saturated zones. It is important to add this information to the introduction to provide the readership with a full background on the usefulness of isotopic tracers for distinguishing sources and processes affecting nitrate in the study area.**
RESPONSE:  Agreed. Added sentences to introduction on the expected d18O from nitrification of organic N sources based on this aquifers' isotope water isotope data.

**Due to the importance of land use changes and the trends away from manure additions towards synthetic fertilizers, is appears highly desirable to describe the changes in agricultural practices at the study site in a bit more detail in this manuscript.**
RESPONSE:  Agreed.  We made changes to the introduction (added Lines 122-125) describing the longer-term general shifts in land use practices.

**The authors have made an excellent effort to constrain their sampling of the aquifer to**

**wells that access aerobic groundwater of average age of less than 5 years to link the detected trends to recent agricultural activities. While this argumentation holds most likely true for the water-saturated portion of the study area, it is important to realize that a similar reasoning is not entirely valid for the unsaturated zone including the soils. In the water-unsaturated and soil zones, "subsurface biogeochemical processes" are certainly ongoing with N immobilization and re-mineralization potentially delaying N transfers for years or decades (see for instance Sebilo et al. (2013): Long-term fate of nitrate fertilizer in agricultural soils; PNAS 110(45): 18185-18189), although the manuscript text on line 134 seems to suggest the opposite. Throughout the manuscript, the authors should make it more clear that their approach provides only very limited insights into N cycling and its transit times in the soil and water-unsaturated zones.**

RESPONSE: While we agree with this comment as a general observation, we must respectfully counter that unsaturated zone biogeochemical reprocessing of N sources is not *necessarily the case and of a significant magnitude everywhere*. As described here and in previous work, this aquifer has thin and poorly developed organic soil and is comprised of coarse sand, gravel and cobbles with poor water retention no capacity to develop anaerobic zone potentials for denitrification. This is evident by fully aerobic conditions nearly everywhere in the aquifer. Moreover, as demonstrated in detailed unsaturated zone isotope tracer experimental work done over this aquifer by Loo et al, they showed that the N isotopic fingerprints of manure versus fertilizer applications were retained in the unsaturated zone, albeit with dampened signals due to N reservoir mixing. If anything, this case study refutes the general idea that source N signals are *always* be masked by unsaturated zone biogeochemical processes. N cycling processes, or lack thereof, are site specific. However, as noted any shifts in sources could indeed take a longer time to be seen due to source changes and vadose zone mixing, which is what we describe (subtle decadal shifts in sources).

**In Figure 2a and associated text on lines 216-224, the authors assign the nitrate isotope data to three nitrate sources. Nitrate in irrigation water (d15N of +9 ‰ and manure (d15N of +8 ‰ are the sources with the highest 15N values, but Figure 2a shows numerous samples with d15N values between 10 and 17 ‰. A short explanation for these elevated d15N values is desirable at this point in the manuscript.**

RESPONSE: We added a sentence to refer to Wassenaar (1995) that showed $d^{15}N$ values up to +14 ‰ in *nitrate* affected by ammonia volatilization and/or nitrification around poultry storage and spreading operations, despite rather limited data. Higher localized $d^{15}N$ values for nitrate derived from poultry manure are likely, but not the manure *solids* themselves.

**Line 241: The mean d15N value of +5.0 ‰ is not very close to that of synthetic fertilizers (d15N near 0 ‰. Is it possible that intensive N cycling in the soil with associated N isotope effects causes a shift to higher d15N values in the seepage water nitrate? If this is a requirement to explain the data patterns, this should be acknowledged in the text of this manuscript.**

RESPONSE: Disagree. We noted that the fertilizers used in the area have a mean d15N of +3.2+/-2.3 as reported in Loo et al (2017). This is consistent with our interpretation.

**In my view, the evidence for climatic impacts on trends in the chemical and isotopic composition of groundwater nitrate presented in this manuscript is very weak (e.g. lines**

**323-325) and is mainly based on references to data presented elsewhere rather than in this manuscript. I am not convinced that a few years on increased precipitation (2008-2011) justify mentioning "climatic changes" in the title of this manuscript especially since no climate data are presented.**
RESPONSE:  Agreed.  We replaced climatic changes with 'precipitation changes'. See the citation given for details.

**Table 1 lists another nitrate source, namely nitrate-containing irrigation water with a ïA˛d'15N of +9 ‰ evidently derived from manure-applications. Throughout the text, this nitrate source receives very little attention. Is it not relevant?**
RESPONSE:  True. But as noted in previous works on the aquifer, groundwater irrigation water (i.e. $^{15}N$ enriched nitrate laden groundwater) is only applied during Jun-Aug during peak ET and maximum crop nutrient uptake, resulting in limited, summer recharge by irrigation water.  The isotope signal is anyhow not readily distinguishable from contemporary manure sources.

In addition, there are a number of minor deficiencies that include the following more specific comments:

**Line 47-50: It should be made clear that atmospheric nitrate inputs are not leached into the groundwater conservatively, but usually undergo intensive recycling via immobilization and ammonification + nitrification in the unsaturated zone prior to reaching the groundwater zone.**
RESPONSE: See above response to vadose zone N recycling comments.

**In line 58, the authors state that manure-derived nitrate has d15N typically >10‰ but subsequently report on line 61 that the d15N of poultry manure in the study area is closer to 8 ‰.  What explains the discrepancy? Is the former range mainly for cattle manure?**
RESPONSE: See above response to Reviewer #1 – we added a sentence about higher d15N values encountered up to +14 per mil observed in nitrates formed around poultry operations.

**In lines 101-107 the aquifer is well described, but one essential piece of information, the depth of the water table below ground surface is not clearly revealed. The authors should add this information in a more transparent fashion;**
RESPONSE: See above response to Reviewer #1.  Water table depth is not a static value. It is highly variable spatially and over time. Instead, Table 4 shows the average water column height in meters as defined at the mid-screen depth below average static water, which is more relevant to the $^3H$-$^3He$ groundwater 'ages' than a depth profile would provide due to large variations in unsaturated zone depth across the aquifer. We believe this is a better way to show this information, and prefer to keep it.

**To support the statement that the aquifer is largely under aerobic conditions it would be beneficial to add dissolved oxygen concentrations to the manuscript (for instance in table 4).**
RESPONSE:  Agreed – also noted by Reviewer #1 – dissolved $O_2$ was added to SM Table.

**In section 2.2, it would be useful to list the depths of water table below ground surface for the 19 selected monitoring wells.**
RESPONSE: See above, same comment.

**Lines 146-7: The measurement uncertainties for concentration analyses (e.g. nitrate, chloride) should be provided;**
RESPONSE: Agreed. Added the MDL.

**Section 3.1: throughout this section it would be more correct to speak about nitrate concentrations of groundwater obtained from wells (since wells have no nitrate concentrations);**
RESPONSE: Agreed. Changed where appropriate.

**Line 178: state by how much the nitrate concentration increased over the 5-year observation period; if you exclude the three wells mentioned on line 184, is there still an increase in nitrate concentrations for the groundwater from the remaining 16 wells?**
RESPONSE: Addressed.

**Line 215: The end-member with low d15N values appears to have "intermediate" nitrate concentrations;**
RESPONSE: Agreed, changed as appropriate.

**Lines 221-222: The rational why the Bayesian clustering model that suggest 5 groupings results in 4 distinct groups (line 223) is not clear to me.**
RESPONSE: Addressed. The first two Bayesian groups were amalgamated as they suggested the same isotopic source, but with partial enrichment.

**Line 236-239: Are these 4 groups shown in any Figures? Also, to which category belong the samples with d15N values between 10 and 17 ‰**
RESPONSE: Addressed. Figure 3 and d15N 9 to 16 is Group 3.

**Line 278: I suspect not the soil N is flushed to the aquifer, but nitrate derived from nitrification of soil N. –**
RESPONSE: See response to vadose zone comment above.

**Line 288: Can you quantify the extent of this decrease in d15N over 5 years? How does it compare to the long-term decrease in groundwater nitrate d15N observed since 1995?**
RESPONSE: Addressed.

**Line 328: Logic unclear: if 14N was preferentially volatilized, should the remaining N compound not be enriched in 15N?**
RESPONSE: Fixed.

**Line 359: Please quantify the extent of the observed decrease in d15N values.**
RESPONSE: Addressed.

The manuscript is written in excellent English, it follows a logical sequence and is hence very well organized, and the objectives are clearly stated. The applied methods are leading-edge and are sufficiently described. Previous literature is exhaustively considered.

Figures and tables are of good quality with minor deficiencies listed below.
Hence, if the authors are able to address the limitations identified in this review, publication of this manuscript after moderate revisions is recommended.

Additional technical comments:

Line Comment

**20 I suspect you did not measure "recharge" directly, but shallow groundwater up to 5 years old; please re-word accordingly;**
**RESPONSE**: Agreed. Changed.

**35 one or two more recent references that are less than 10 years old may be desirable;**
**RESPONSE**: Agreed. Newer citations added.

**43 do you mean "nitrate" isotopes or also other isotopic parameters? If the latter, please mention which other isotopes?**
**RESPONSE**: Nitrate. Re-worded in the paragraph.

**53 are these d15N values representative for this study site?**
**RESPONSE**: Yes. As cited in Loo et al.

**62 & following I suspect the numbers in brackets are N-P-K values for synthetic fertilizers, but this may need to be explained to the readership.**
**RESPONSE**: Corrected.

**65 add a reference to support this statement;**
**RESPONSE**: Reference added.

**67 it would be advantageous to spell out the fertilizer sources used in the study area;**
**RESPONSE**: Corrected.

**70 a more detailed explanation on how the oxygen isotope ratios of nitrate derived from nitrification are controlled is needed here;**
**RESPONSE**: Added sentence to clarify.

**75 samplings (should likely be plural)**
 **RESPONSE**: Corrected.

**80 add a reference for "winter-biased recharge";**
**RESPONSE**: Corrected.

**84 . . . with a focus on "shallow groundwater" from water table wells . . .**
**RESPONSE**: Corrected.

**86 something seems wrong or duplicated here: ..." isotope nitrate and isotope . . ."; also "processes" should be plural;**
**RESPONSE**: Corrected.

**100 "unpublished data" should be moved inside the brackets;**
**RESPONSE**: Corrected.

**105 delete "surface"**
**RESPONSE**: Corrected.

**119 . . . of nitrate "in groundwater" . . .**
**RESPONSE**: Corrected.

**121 if possible add average depths for deep wells and average nitrate-N concentrations;**
**RESPONSE**: Corrected.

**132 not the wells are aerobic, but the groundwater obtained from the wells;**
**RESPONSE**: Corrected.

**157 in d18O the "1" appears to be missing;**
**RESPONSE**: It is there.

**168 does this refer to "nitrate concentrations"?**
**RESPONSE**: Corrected.

**193 indicate in which months the first major recharge occurs? Is it late fall?**
**RESPONSE**: Clarified.

**197 are vadose zone infiltration lag-times similar for all sites?**
**RESPONSE**: Corrected.

**199 the groundwater is aerobic, not the wells; I did not find a supplementary table;**
**RESPONSE**: Wording corrected and added mean DO concentration. Added DO to the SM table.

**200 delete "a"**
**RESPONSE**: Corrected.

**206 what does this limited variability indicate? Longer transit times through the unsaturated zone?**
**RESPONSE**: addressed.

**208 throughout this section, the d15N values are for nitrate in groundwater, not for wells.**

**RESPONSE**: Corrected.

**210, 213, 214 no need to report data with 2 decimal places given the measurement uncertainty of this parameter;**
**RESPONSE**: Corrected.

**250 what is meant with "like group 1a values". d15N of 6.7 ‰ is not like 5.0 ‰ and even further away from synthetic fertilizer d15N values of 0 ‰**
**RESPONSE**: Corrected.

**261 groundwater flow paths are neither shown in Figure 1 nor in 4;**
**RESPONSE**: Corrected.

**270 . . . influenced "the nitrate contamination level" in these wells;**
**RESPONSE**: Corrected.

**278 replace "isotopically" with 15N-enriched;**
**RESPONSE**: Corrected.

**297 could you not have microbial transformations but with negligible isotope fractionation?Almost all transformations in the N cycle are microbially mediated;**
**RESPONSE**: Yes but with continuous input of groundwater N, resulting in no appreciable buildup of enriched N. But our N inputs are seasonal, driven by winter recharge.

**305 I could not find the supplementary table;**
**RESPONSE**: It is there.

**314 it is not possible to enrich a delta value. Also, do you mean enrichment in 14-N or 15-N?**
**RESPONSE**: Corrected

**323 increasing trend for which parameter?**
**RESPONSE**: 15N. Corrected

**334 Wassenaar et al. (2006)**
**RESPONSE**: Corrected.

**336 d18O of nitrate**
**RESPONSE**: Corrected.

**338 were anaerobic conditions detected based on DO concentrations? If so what were the DO concentration ranges?**
**RESPONSE**: Corrected and added to SM as per other reviewers.

**342 and Wassenaar et al. (2006)**
**RESPONSE**: Corrected

**343 depletion of 15N in what: in groundwater nitrate? Why 15-N depletion if you previously talked about denitrification?**
RESPONSE: Corrected.

**363-4 this is new information that was not previously provided in the Results & Discussion section.**
RESPONSE: It is evident in the maps presented beforehand. Added text to refer to this map.

**376 do you men enrichment in 14-N or 15-N of nitrate?**
RESPONSE: Fixed.

**373 do you mean concentrations and isotopic compositions of nitrate?**
RESPONSE: Corrected

**390 do you mean "groundwater" nitrate?**
RESPONSE: Corrected

**518 the inset map requires a distance bar (in km);**
RESPONSE: Corrected

**541 units are missing for d15N**
RESPONSE: Corrected

**544 Table 2: why are concentrations listed here as nitrate, when throughout the rest of the manuscript they are given as nitrate-N? Also units are missing.**
RESPONSE: Corrected

**547 Depletion (rather than depleting)?**
RESPONSE: Corrected

**550 nitrate concentration unit is wrong: mg/L rather than ‰**
RESPONSE: Corrected